# Tackling Feature-Classifier Mismatch in Federated Learning via Prompt-Driven Feature Transformation

**Xinghao Wu**[1][*], **Xuefeng Liu**[1,5], **Jianwei Niu**[1,2,5][†], **Guogang Zhu**[1],

**Mingjia Shi**[6], **Shaojie Tang**[3], **Jing Yuan**[4],

[1]State Key Laboratory of Virtual Reality Technology and Systems,
School of Computer Science and Engineering, Beihang University, Beijing, China

[2]Hangzhou Innovation Institute of Beihang University, Zhejiang Key Laboratory of
Industrial Big Data and Robot Intelligent Systems, Hangzhou, China

[3] Center for AI Business Innovation, Department of Management Science and Systems,
University at Buffalo, Buffalo, New York, USA

[4] University of North Texas, Denton, Texas, USA    [5] Zhongguancun Laboratory, Beijing, China

[6] Sichuan University, Sichuan, China

## Abstract

Federated Learning (FL) faces challenges due to data heterogeneity, which limits the global model's performance across diverse client distributions. Personalized Federated Learning (PFL) addresses this by enabling each client to possess an individual model adapted to its local distribution. Many existing methods assume that certain global model parameters are difficult to train effectively in a collaborative manner under heterogeneous data. Consequently, they localize or fine-tune these parameters to obtain personalized models. In this paper, we reveal that both the feature extractor and classifier of the global model are inherently strong, and the primary cause of its suboptimal performance is the mismatch between local features and the global classifier. Although existing methods alleviate this mismatch to some extent and improve performance, we find that they either (1) fail to fully resolve the mismatch while degrading the feature extractor, or (2) address the mismatch only post-training, allowing it to persist during training. This increases inter-client gradient divergence, hinders model aggregation, and ultimately leaves the feature extractor suboptimal for client data. To address this issue, we propose FedPFT, a novel framework that resolves the mismatch during training using personalized prompts. These prompts, along with local features, are processed by a shared self-attention-based transformation module, ensuring alignment with the global classifier. Additionally, this prompt-driven approach offers strong flexibility, enabling task-specific prompts to incorporate additional training objectives (*e.g.*, contrastive learning) to further enhance the feature extractor. Extensive experiments show that FedPFT outperforms state-of-the-art methods by up to 5.07%, with further gains of up to 7.08% when collaborative contrastive learning is incorporated. The code is available at https://github.com/XinghaoWu/FedPFT.

## 1 Introduction

Federated Learning (FL) [34, 65, 24, 21, 23, 22] enables clients to collaboratively train a global model without sharing their raw data. A major challenge in FL is data heterogeneity, where data

---

[*]Email: wuxinghao@buaa.edu.cn

[†]Corresponding author: niujianwei@buaa.edu.cn

39th Conference on Neural Information Processing Systems (NeurIPS 2025).

| Methods | CIFAR-10, $\alpha = 0.5$ | | | CIFAR-10, $\alpha = 1.0$ | | |
|---|---|---|---|---|---|---|
| | Match Acc. | Origin Acc. | Probe Acc. | Match Acc. | Origin Acc. | Probe Acc. |
| FedAvg | 70.71% | 60.05% | 71.36% | 67.34% | 60.90% | 67.87% |
| FedBABU | 71.86% | 60.87% | 71.33% | 68.55% | 62.96% | 68.27% |
| FedPer | 70.23% | 69.50% | 70.20% | 66.42% | 65.92% | 66.12% |
| FedBN | 69.42% | 66.19% | 69.80% | 65.34% | 63.39% | 65.61% |
| FedCAC | 70.26% | 68.15% | 70.52% | 67.22% | 65.77% | 67.47% |
| FedAvg-FT | 71.21% | 70.63% | 71.36% | 66.84% | 66.44% | 67.87% |
| FedBABU-FT | 72.01% | 71.69% | 71.33% | 68.97% | 68.42% | 68.27% |
| FedPFT | 72.81% | 72.66% | 71.91% | 69.26% | 69.08% | 69.38% |
| FedPFT+Con | 76.71% | 76.83% | 76.80% | 73.52% | 73.15% | 74.50% |

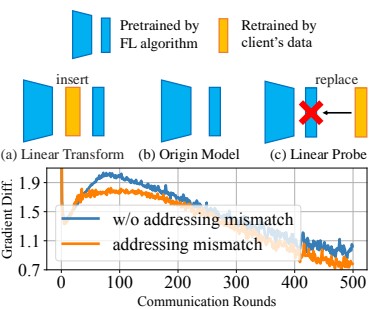

Figure 1: ***Left:*** Match Acc. represents the accuracy after applying a linear transformation to the features to adapt them to the classifier. Origin Acc. indicates the accuracy of the original model. Probe Acc. refers to the accuracy achieved by retraining the classifier with local data. All accuracies are obtained on the client testing data. **The disparity between Origin Acc. and Match Acc. indicates the degree of mismatch** (see Appendix A for a formal definition). ***Right-Top:*** (a), (b), and (c) are toy examples of the models used to calculate the three types of accuracy. ***Right-Bottom:*** Addressing mismatch (orange line) during training reduces inter-client gradient divergence compared to not addressing it (blue line).

distributions across clients are not independently and identically distributed (non-IID). Under such conditions, a single global model often fails to generalize well across all clients, leading to suboptimal local performance.

To address this, Personalized Federated Learning (PFL) has been proposed, enabling each client to maintain an individual model tailored to its local data distribution. A predominant approach in PFL involves generating the personalized model by localizing or fine-tuning a selected subset of the global model parameters, based on the premise that these parameters are inherently challenging to optimize effectively through collaborative training. For instance, FedPer [1] personalizes the classifier, while FedBABU [37] fine-tunes it. FedBN [26] introduces personalized batch normalization (BN) layers, and methods like AlignFed [63, 64] personalizes feature extractors. Additionally, advanced approaches have adopted more sophisticated techniques, including hypernetwork [33], reinforcement learning [43], or quantitative metrics [50], to select parameters for personalization.

The above research in PFL reveals an intriguing phenomenon: regardless of which components of the global model are personalized (*e.g.*, feature extractors, classifiers, or intermediate layers), various PFL methods tend to achieve comparable performance levels. This observation leads to a non-trivial hypothesis: **each individual component within the global model may already be sufficiently optimized** through conventional FL processes, whereas **the suboptimal performance stems primarily from the mismatch between these well-trained modules**.

To validate this hypothesis, we conduct a linear transform experiment by inserting a simple linear layer between the FedAvg-trained global feature extractor and its classifier, then retraining this layer on each client's local data. As reported in Fig. 1 ("Match Acc."), this adjustment greatly improves accuracy over the original global model ("Origin Acc."). This result reveals that **FedAvg already learns a strong feature extractor and classifier, but their direct combination leads to a feature-classifier mismatch when applied to client-specific data**. A toy example illustrating this mismatch is shown in Fig. 2.

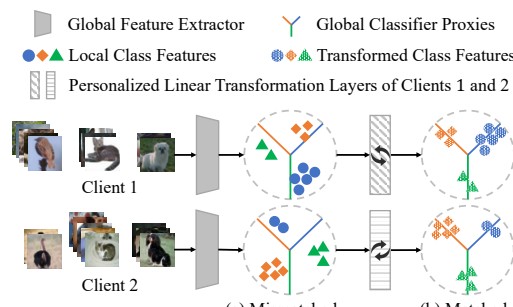

Figure 2: A toy example illustrating the mismatch in FedAvg and how a linear transform addresses it. (a) The local features from two clients have well-clustered structures, but due to the effects of non-IID data, their features mismatch with the global classifier proxies. (b) By applying personalized transformations to the local features of each client, the features are aligned with the global classifier.

Notably, existing PFL methods that localize (lines 5-7) or fine-tune (lines 8-9) only part of the model parameters mitigate this mismatch, which explains their strong performance despite modifying a small portion of the model. However, these personalization strategies come with important limitations. Our linear probe experiments in Fig. 1 indicate that localizing partial parameters degrades the quality of the global feature extractor (*i.e.*, lowering the "Probe Acc."), thereby lowering the model's upper performance bound.

In contrast, fine-tuning the classifier preserves the strong feature representations learned by the global feature extractor, leading to higher "Match Acc.". However, it remains a **post-training adjustment**, addressing mismatch only after the global model has been trained. Since mismatch persists throughout training, clients must further adjust the feature extractor to align with the classifier in each round, exacerbating inter-client model divergence. This leads to detrimental interference during model aggregation, making the global feature extractor suboptimal for client data. As illustrated in Fig. 1 (right-bottom), when the mismatch exists (blue line), the gradient divergence among client models is larger.

To resolve the mismatch problem **throughout training**, we introduce a novel PFL method, FedPFT. While inserting a personalized linear layer between the shared feature extractor and shared classifier is a straightforward approach, it can easily overfit the limited local training data due to the large number of personalized parameters, and it is insufficient for handling complex datasets (as demonstrated in the Appendix H). To overcome these limitations, we draw inspiration from prompt technology [11], which uses prompts to guide model behavior. FedPFT incorporates personalized prompts with minimal trainable parameters and a shared self-attention-based feature transformation module (FTM). Both the prompts and local features are fed into the FTM, where features are transformed via the attention mechanism. In each round, FedPFT first trains the prompts to align local features with the global classifier. Subsequently, training the model parameters based on this alignment reduces the model divergence among clients and enhances the synergy between the feature extractor and classifier. The results in Fig. 1 demonstrate that FedPFT not only resolves the mismatch problem but also improves the feature extractor.

Another advantage of our designed prompt-driven FTM is its strong flexibility across different tasks. It can leverage task-specific prompts to incorporate various tasks beneficial for client collaboration, such as contrastive learning [20, 48], feature alignment [61, 54], etc. Taking contrastive learning as an example, as shown in Fig. 1, FedPFT+Con further improves model performance by introducing collaborative contrastive learning through prompts.

Our main contributions can be summarized as follows:

- We identify that the global model's inadequate performance on client local data in non-IID scenarios is primarily due to the mismatch between local features and the classifier. We show that the reason localizing or fine-tuning some parameters improves performance is that it indirectly alleviates this issue. This insight offers a new perspective for future PFL approaches to better address the non-IID problem.
- We propose a new PFL framework that incorporates a prompt-driven feature transformation module to align local features with the global classifier. This approach not only resolves the mismatch problem but also provides flexibility for incorporating various collaborative tasks to further enhance PFL performance.
- Our experiments on multiple datasets and non-IID scenarios (including both label shift and feature shift) demonstrate the superiority of FedPFT, outperforming state-of-the-art (SOTA) methods by up to 5.07%. When further incorporating contrastive learning task, this improvement can reach up to 7.08%.

## 2   Related Work

PFL [45, 60] has emerged as an effective approach to addressing the challenges posed by non-IID data in FL. In recent years, numerous methodologies have been proposed in this domain [6, 25, 31, 46, 59, 57, 52, 62, 40, 39, 35]. In this section, we focus on three categories of methods most relevant to our work.

**Parameter decoupling methods** aim to decouple a subset of parameters from the global model for personalization. Approaches such as FedPer [1], FedRep [4], and GPFL [58] focus on personalizing

the classifier. In contrast, methods like LG-FedAvg [28] and AlignFed [64] advocate for the personalization of the feature extractor. Additionally, FedBN [26] and MTFL [36] propose personalizing batch normalization (BN) layers within the feature extractor. Techniques employing deep reinforcement learning [43] or hypernetworks [33] have been used to determine which specific layers to personalize. The recent FedCAC [50] and FedSelect [44] further advance this by enabling parameter-wise selection. FedDecomp [49] extends this idea by decomposing each parameter into a personalized component and a shared component, achieving finer-grained decoupling. These decoupling methods indirectly help alleviate the mismatch issue within the global model by allowing local parameter adjustments. However, these methods do not completely resolve the mismatch issue. Furthermore, excessive parameter personalization reduces information exchange across clients, which can degrade the overall quality of the feature extractor.

**Classifier fine-tuning-based methods** such as FedBABU [37] and FedETF [27] propose that when the global model is well-trained, fine-tuning the classifier alone can yield strong personalized performance. Consequently, these methods adopt a two-stage training strategy: in the first stage, they focus on training a high-quality global model, and in the second stage, they fine-tune the classifier to obtain a personalized model. By improving the feature extractor in the first stage, these methods provide a stronger foundation for addressing the mismatch through classifier fine-tuning, leading to better personalized performance. However, since the mismatch persists throughout the global model training process, clients must continuously adjust the model to align the feature extractor and classifier in each round. This exacerbates inter-client model divergence, causing interference during aggregation and ultimately making the feature extractor suboptimal for client data.

**Feature alignment methods** have shown that non-IID data can lead to discrepancies in feature distributions across clients, which in turn cause client drift and hinder model training. To mitigate this issue, existing approaches align features across clients, for example, through global prototype alignment [54, 61, 53] or classifier calibration [32, 7, 56, 37], thereby reducing inter-client feature divergence. However, these methods focus solely on inter-client alignment and overlook the feature–classifier mismatch that occurs within each client, which we identify as a key bottleneck. In contrast, our approach directly addresses this mismatch by aligning local features with the global classifier, which also implicitly promotes inter-client feature alignment through a shared semantic space.

**Prompt-tuning-based methods.**   Prompt tuning has been widely adopted in vision and language tasks as a lightweight way to adapt pre-trained models [11, 29, 16, 30]. Recent works such as pFedPG [55], SGPT [5], FedOTP [19], FedAPT [42], and pFedPT [18] extend this idea to PFL, where prompts are optimized to adapt a shared, well-trained backbone to local data. These works primarily focus on *how to adapt a well-trained global model to fit local client distributions*. In contrast, our approach addresses a fundamentally different problem: *how to mitigate feature–classifier mismatch during the training of the backbone itself*. Rather than applying prompts as a fine-tuning tool, we incorporate them as a mechanism to improve alignment and reduce gradient divergence throughout training. Our experiments (see Section 4.4) further demonstrate that post-training prompt tuning is insufficient to match the performance of our training-time alignment strategy.

## 3   Methodology

### 3.1   FedPFT Framework: Architecture and Workflow Overview

As illustrated in Fig. 3(a), the core of FedPFT is the introduction of a prompt-driven feature transformation module (FTM) $\tau_i$ between the feature extractor $\phi_i$ and task-specific heads (*e.g.*, classifier). This module transforms local features during training to better align them with their corresponding tasks. As shown in Fig. 3(b), prompts $p$ and image features $f$ are fed into the FTM, where self-attention operations transform $f$ to $f'$, which is then used by task heads. Specifically, FedPFT employs classification prompts $p_{\kappa,i}$ to transform $f$, ensuring alignment with the classifier $h_{\kappa,i}$.

Each training round of each client $i$ in FedPFT includes five key steps:

1. Client downloads the global models, including feature extractor $\phi$, FTM $\tau$, and classifier $h_\kappa$.
2. Client freezes the global feature extractor $\phi$ and updates $\tau_i$ and prompts $p_{\kappa,i}$ using the cross-entropy loss $L_{\text{CE}}$ to align local features with the frozen global classifier $h_\kappa$.

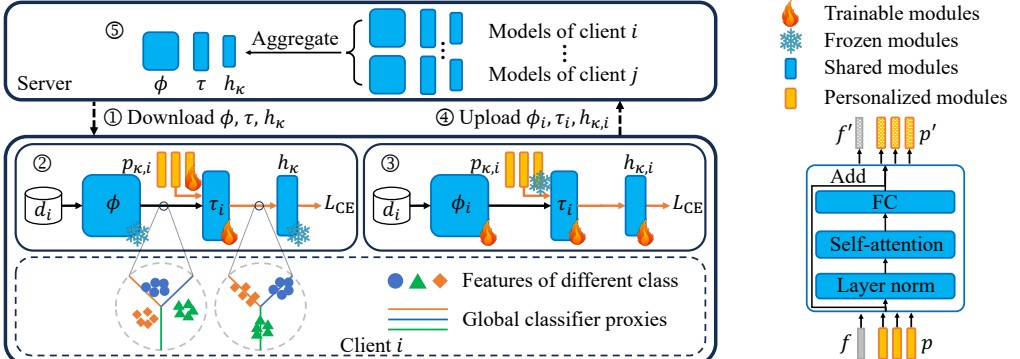

(a) The overview of FedPFT in one communication round     (b) The feature transformation module ($\tau$)

Figure 3: Overview of FedPFT. (a) The training process of each client $i$ in one communication round. (b) The feature transformation module in FedPFT.

3. Based on the alignment between the features and the classifier, the client freezes the prompts $p_{\kappa,i}$ and updates $\phi_i, \tau_i, h_{\kappa,i}$ with $L_{CE}$ to learn client local knowledge.

4. Client uploads $\{\phi_i, \tau_i, h_{\kappa,i}\}$ to the server while retaining $\{p_{\kappa,i}\}$ locally.

5. The server aggregates the models uploaded by the clients.

## 3.2 Problem Formulation

In PFL, $N$ clients train their personalized models $w_i, i \in [N]$ under the coordination of a server, aiming for each $w_i$ to perform well on client data distribution $\mathbb{D}_i$. This objective can be formalized as $\min_{\{w_i\}_{i\in[N]}} \frac{1}{N} \sum_{i=1}^{N} L_i(w_i; \mathbb{D}_i)$, where $L_i$ represents the loss function of the $i$-th client.

In this work, we aim to enhance personalized models by addressing the mismatch problem between local features and the classifier in the global model. The training objective of FedPFT is thus formulated as:

$$\min_{\phi,\tau,h_\kappa} \min_{\{p_{\kappa,i}\}_{i\in[N]}} \mathbf{E}_i\{L_i(\phi, \tau, h_\kappa, p_{\kappa,i}; d_i) := \mathbf{E}_{d_i}[L_{CE}(\phi, \tau, h_\kappa, p_{\kappa,i}; d_i)]\}, \tag{1}$$

where $\phi$ and $h_\kappa$ represent the global feature extractor and global classifier, respectively. $\tau$ is the newly introduced global FTM. This module, along with the personalized classification prompt $p_{\kappa,i}$, transforms local features to align with the global classifier. $L_{CE}$ denotes the cross-entropy loss for classification tasks. $d_i$ represents the dataset of the client.

## 3.3 Feature Transformation Module (FTM)

In FedPFT, given a sample $x_j \in d_i$, processed by the feature extractor $\phi$, the extracted feature is $f_j \in \mathbb{R}^m$, where $m$ is the feature dimension. A collection of $n$ prompts is denoted as $p = \{\boldsymbol{p}^k \in \mathbb{R}^m | k \in \mathbb{N}, 1 \leq k \leq n\}$. The FTM operates as follows:

$$[f'_j, p'] = \tau([f_j, p]), \tag{2}$$

where $[\cdot, \cdot]$ signifies stacking and concatenation along the sequence length dimension, yielding $[f'_j, p'] \in \mathbb{R}^{(1+n) \times m}$. Here, the $f'_j \in \mathbb{R}^m$ represents the transformed feature, which is subsequently used for downstream tasks. An illustration of the FTM is provided in Fig. 3(b). In FedPFT, we denote $n_\kappa$ as the number of prompts contained in $p_{\kappa,i}$.

The FTM dynamically adapts local features to downstream tasks, offering strong flexibility. By employing task-specific prompts $p$, it can introduce auxiliary objectives beneficial for client collaboration. We explore this further in Section 3.6.

## 3.4 Classification Task with Personalized Prompts

FedPFT employs a set of personalized prompts $p_{\kappa,i}$ as inputs to the FTM, enabling each client $i$ to transform its local features for better alignment with the global classifier. Specifically, the

classification loss for client $i$ is defined as:

$$L_{\text{CE}}(\phi, \tau, p_{\kappa,i}, h_\kappa; x, y) = -\log \sum_{c=1}^{C} y_c \log(o_{i,c}), \text{where } x, y \sim d_i. \qquad (3)$$

$C$ is the number of classes, and $o_i = \text{Softmax}(h_\kappa \circ \tau([\phi(x), p_{\kappa,i}]))$ represents the predicted probabilities, with $o_{i,c}$ being the ones of class $c$.

### 3.5 Dual-Phase Alternating Optimization Strategy

To effectively resolve the mismatch problem and coordinate the training of different modules in FedPFT, we propose an alternating optimization strategy, which partitions each local training round into two phases: the feature transformation phase and the model training phase.

**Feature transformation phase.** In this phase, the objective is to align local features with the classifier by optimizing the classification prompts $p_{\kappa,i}$ and the FTM $\tau_i$ while keeping the global feature extractor $\phi$ and global classifier $h_\kappa$ frozen. The training objective is formulated as:

$$\min_{p_{\kappa,i}, \tau_i} L_{\text{CE}}(\tau_i, p_{\kappa,i}; \phi, h_\kappa, d_i). \qquad (4)$$

**Model training phase.** Following the above phase, this phase focuses on learning client-specific knowledge by updating the feature extractor, FTM, and classifier while keeping the prompts fixed. The training objective is given by:

$$\min_{\phi_i, \tau_i, h_{\kappa,i}} L_{\text{CE}}(\phi_i, \tau_i, h_{\kappa,i}; p_{\kappa,i}, d_i). \qquad (5)$$

Let $R$ represent the total number of local epochs per training round. We divide it into $R_f$ epochs for the feature transformation phase and $R_a$ epochs for the model training phase, satisfying $R_f + R_a = R$. It is crucial that $R_f$ is always larger than $R_a$ to ensure that the mismatch between local features and the classifier is resolved before training the model parameters.

After completing local training, the updated parameters $\phi_i$, $\tau_i$, and $h_{\kappa,i}$ are aggregated at the server to facilitate client collaboration, while the personalized prompts $p_{\kappa,i}$ remain local. We simply adopt the aggregation method used in FedAvg. The pseudo-code of FedPFT is summarized in the Appendix D.

### 3.6 FedPFT with Additional Tasks: An Example of Contrastive Learning

As discussed in Section 3.3, our FTM provides strong flexibility. Benefiting from this, the FedPFT framework can seamlessly integrate additional tasks beneficial for PFL, such as contrastive learning [20, 48], feature alignment [61, 54], multi-task learning, etc., by simply incorporating task-specific prompts. In this section, we utilize contrastive learning as an example.

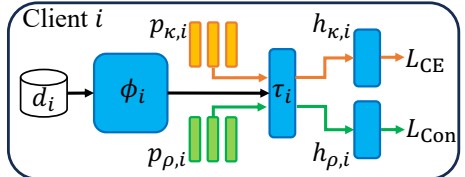

Figure 4: FedPFT with contrastive learning. The orange solid line and the green solid line represent the forward propagation of the classification task and the contrastive learning task, respectively.

As depicted in Fig. 4, we introduce another set of personalized prompts $p_{\rho,i}$, which are fed into $\tau_i$ to transform features for the contrastive learning task with a global projection head $h_{\rho,i}$. The goal is to enable all clients to collaboratively optimize the contrastive learning loss, thereby improving the performance of the feature extractor. The training objective can be formulated as

$$\min_{\phi, \tau, h_\kappa} \min_{\{p_{\kappa,i}\}_{i \in [N]}} \mathbf{E}_i \{L_i(\phi, \tau, h_\kappa, p_{\kappa,i}; d_i) := \mathbf{E}_{d_i}[L_{\text{CE}}(\phi, \tau, h_\kappa, p_{\kappa,i}; d_i) + L_{\text{Con}}(\phi, \tau; d_i)]\}, \quad (6)$$

where $L_{\text{Con}}$ represents the contrastive learning loss function. The detailed optimization process and the definition of $L_{\text{Con}}$ are provided in the Appendix E. In Section 4, we refer to this extended method as FedPFT+Con, validating the effectiveness of combining FedPFT with contrastive learning.

Table 1: Test accuracy (%) of different methods under Dirichlet non-IID scenarios on CIFAR-10, CIFAR-100 and Tiny ImageNet. The best results are highlighted as first , second , and third .

| | CIFAR-10 | | | CIFAR-100 | | | Tiny ImageNet | | |
|---|---|---|---|---|---|---|---|---|---|
| Methods | $\alpha=0.1$ | $\alpha=0.5$ | $\alpha=1.0$ | $\alpha=0.1$ | $\alpha=0.5$ | $\alpha=1.0$ | $\alpha=0.1$ | $\alpha=0.5$ | $\alpha=1.0$ |
| FedAvg | 56.81±0.07 | 60.73±1.85 | 61.53±0.93 | 34.47±0.85 | 31.96±0.70 | 33.20±0.62 | 19.15±1.15 | 18.38±0.48 | 17.73±0.73 |
| FedBABU | 55.87±1.68 | 61.44±0.92 | 63.57±1.36 | 36.37±0.76 | 33.02±1.15 | 33.61±0.35 | 20.05±1.29 | 19.07±1.14 | 17.77±0.37 |
| FedETF | 56.27±0.64 | 59.62±1.22 | 61.51±1.40 | 33.84±1.05 | 31.10±1.05 | 32.24±0.24 | 17.96±0.54 | 15.50±0.80 | 15.18±0.33 |
| FedFA | 56.93±0.73 | 61.02±1.98 | 62.51±0.88 | 35.16±1.08 | 34.07±0.78 | 34.69±1.58 | 15.86±0.95 | 17.49±0.61 | 16.75±0.60 |
| Local | 81.91±3.09 | 60.15±0.86 | 52.24±0.41 | 47.61±0.96 | 22.65±0.51 | 18.76±0.63 | 24.07±0.62 | 8.75±0.30 | 6.87±0.28 |
| FedAvg-FT | 85.55±2.21 | 72.00±2.05 | 67.78±0.67 | 60.08±0.90 | 40.72±0.55 | 37.54±0.46 | 40.36±1.13 | 25.11±0.33 | 21.33±0.59 |
| FedBABU-FT | 85.57±1.78 | 72.16±1.34 | 68.43±1.20 | 60.10±0.54 | 40.95±0.92 | 38.05±0.89 | 40.62±1.32 | 25.47±1.04 | 21.37±0.51 |
| FedETF-Per | 83.49±2.31 | 67.90±2.58 | 63.51±0.49 | 57.00±0.92 | 36.87±1.97 | 35.18±1.32 | 34.04±0.66 | 19.98±1.68 | 15.44±0.28 |
| FedFA-FT | 80.70±2.98 | 67.68±2.26 | 66.50±0.48 | 42.62±0.59 | 33.39±0.41 | 33.88±0.79 | 24.03±0.61 | 18.07±0.57 | 16.51±0.50 |
| FedAMP | 84.99±1.82 | 68.26±0.79 | 64.87±0.95 | 46.68±1.06 | 24.74±0.58 | 18.22±0.41 | 27.85±0.71 | 10.70±0.32 | 7.13±0.21 |
| FedPer | 84.43±0.47 | 68.80±0.49 | 64.92±0.66 | 51.38±0.94 | 28.25±1.03 | 21.53±0.50 | 32.33±0.31 | 12.69±0.42 | 8.67±0.40 |
| FedRep | 84.59±1.58 | 67.69±0.86 | 60.52±0.72 | 51.25±1.37 | 26.97±0.33 | 20.63±0.42 | 30.83±1.05 | 12.14±0.28 | 8.37±0.25 |
| FedBN | 83.55±2.32 | 66.79±1.08 | 62.20±0.67 | 54.35±0.63 | 36.94±0.94 | 33.67±0.12 | 33.34±0.71 | 19.61±0.35 | 16.57±0.44 |
| FedRoD | 86.23±2.12 | 72.34±1.77 | 68.45±1.94 | 60.17±0.48 | 39.88±1.18 | 36.80±0.56 | 41.06±0.77 | 25.63±1.11 | 22.32±1.13 |
| pFedSD | 86.34±2.61 | 71.97±2.07 | 67.21±1.89 | 54.14±0.77 | 41.06±0.83 | 38.27±0.20 | 39.31±0.19 | 19.25±1.80 | 15.91±0.33 |
| pFedGate | 87.25±1.91 | 71.98±1.61 | 67.85±0.87 | 48.54±0.39 | 27.47±0.79 | 22.98±0.03 | 37.59±0.39 | 24.09±0.67 | 19.69±0.14 |
| FedCAC | 86.82±1.18 | 69.83±0.46 | 65.39±0.51 | 57.22±1.52 | 38.64±0.63 | 32.59±0.32 | 40.19±1.20 | 23.70±0.28 | 18.58±0.62 |
| pFedPT | 82.38±2.91 | 67.33±1.33 | 64.37±1.22 | 43.21±1.66 | 35.23±0.87 | 36.25±0.37 | 23.55±0.68 | 22.35±0.49 | 21.69±0.24 |
| FedPFT | 87.23±2.69 | 74.10±1.95 | 69.23±0.76 | 60.98±0.39 | 44.87±0.76 | 41.83±0.37 | 41.49±0.10 | 28.61±0.40 | 25.10±0.59 |

# 4 Experiments

## 4.1 Experimental Setup

**Datasets.** In this section, we mainly verify FedPFT in the label shift non-IID scenario, which is one of the most commonly used scenarios in FL research. Specifically, we examine two settings: Dirichlet non-IID and Pathological non-IID. In each setting, we employ three datasets: CIFAR-10 [14], CIFAR-100 [13], and Tiny ImageNet [15]. Additionally, in Section 4.6. we also verify FedPFT in the feature shift non-IID scenario with PACS [17] and DomainNet [38] datasets.

In the label shift scenario, each client is assigned 500 training samples. For CIFAR-10 and CIFAR-100 datasets, each client has 100 test samples; for the Tiny ImageNet dataset, each client has 200 test samples. Both training and test data have the same label distribution.

**Baseline methods.** We compare our method against 17 state-of-the-art (SOTA) methods: FedAvg [34], FedBABU [37], FedETF [27], FedFA [61], FedAvg-FT, FedBABU-FT, FedETF-Per, FedFA-FT, FedAMP [10], FedPer [1], FedRep [4], FedBN [26], FedRoD [3], pFedSD [12], pFedGate [2], FedCAC [50], and pFedPT [18]. Here, FedAvg-FT, FedBABU-FT, and FedFA-FT indicate local fine-tuning the classifier of the global model, as we find that fine-tuning the classifier consistently outperforms full-model fine-tuning. FedETF-Per uses the official fine-tuning method to obtain personalized models.

**Hyperparameter settings.** For the general hyperparameters of FL, we set the number of clients $N = 40$, batch size $B = 100$, and local update rounds $R = 5$. The total rounds $T$ are set to 1000 to ensure all methods reach full convergence. We select the highest average accuracy achieved by all clients across all rounds as the result. Each experiment is repeated with three random seeds, and the mean and standard deviation are reported. We employ the ResNet [9] model architecture, specifically ResNet-8 for CIFAR-10 and ResNet-10 for CIFAR-100 and Tiny ImageNet. Please refer to the Appendix F for more details.

## 4.2 Comparison with SOTA Methods

We compare our proposed FedPFT against 17 SOTA methods across three datasets in Dirichlet non-IID scenarios. The experimental results on CIFAR-10, CIFAR-100, and Tiny ImageNet are presented in Table 1. Results in Pathological non-IID scenarios are provided in the Appendix G.

**Results in Dirichlet non-IID scenario.** In this setting, we evaluate the impact of varying non-IID degrees by adjusting $\alpha$. The results, as detailed in Table 1, demonstrate that performance varies significantly depending on the underlying design principles of each method. Among all methods,

FedRoD demonstrates robust performance across all datasets and non-IID degrees. This is primarily due to its dual-classifier design: a personalized classifier for local feature alignment and a global classifier for assistance from other clients to improve generalization. *FedPFT explicitly addresses the mismatch problem and achieves superior results across all scenarios.*

To further validate the benefits of incorporating additional tasks into FedPFT, we also evaluate FedPFT+Con. As shown in Table 2, FedPFT+Con *significantly outperforms existing SOTA methods by up to 7.08%* by integrating contrastive learning into FedPFT. This improvement highlights the strong flexibility of our proposed FTM in enhancing PFL performance.

Table 2: Test accuracy (%) of FedPFT+Con under Dirichlet non-IID on CIFAR-100 and Tiny ImageNet.

| Datasets | $\alpha = 0.1$ | $\alpha = 0.5$ | $\alpha = 1.0$ |
|---|---|---|---|
| CIFAR-10 | 88.60±2.19 | 77.54±1.88 | 74.81±0.77 |
| CIFAR-100 | 62.03±1.41 | 47.98±0.78 | 44.29±0.74 |
| Tiny | 43.42±1.62 | 32.44±0.58 | 27.84±0.41 |

## 4.3 Ablation Study

In this section, we validate the effectiveness of each component in FedPFT on the CIFAR-100 dataset. The experimental results are presented in Table 3.

Setting I represents FedAvg. Setting II incorporates classification prompts $p_\kappa$ to allow each client to adjust the global model individually to obtain a personalized model, resulting in a performance improvement. Setting III (*i.e.*, FedPFT) incorporates alternating training, where prompts are first updated to align local features with the global classifier to address the mismatch problem, followed by training model parameters. This approach effectively resolves the mismatch issue and enhances

Table 3: Experiments on the CIFAR-100 with $\alpha = 0.1$ to illustrate the effectiveness of different modules.

| Settings | $p_\kappa$ | Alter. | $L_{\text{Con}}$ | $p_\rho$ | Accuracy (%) |
|---|---|---|---|---|---|
| I | | | | | 33.87±1.35 |
| II | ✓ | | | | 40.97±1.28 |
| III | ✓ | ✓ | | | 60.98±0.39 |
| IV | ✓ | ✓ | ✓ | | 61.13±0.50 |
| V | ✓ | ✓ | ✓ | ✓ | 62.03±1.41 |
| VI | ✓ | | ✓ | ✓ | 53.76±0.35 |

the synergy between the feature extractor and classifier, *resulting in the greatest performance improvement (20.01%) compared to other modules.*

Setting IV adds contrastive learning loss to Setting III, aiming to further enhance the feature extractor's performance through contrastive learning techniques. Setting V (*i.e.*, FedPFT+Con) incorporates specific prompts $p_\rho$ to better transform features for the contrastive learning task, reducing mutual interference between the two tasks during training.

Setting VI represents incorporating contrastive learning into PFL without addressing the mismatch issue. As shown, while contrastive learning can improve model accuracy by enhancing the quality of the feature extractor, its performance is far inferior to FedPFT (*i.e.*, Setting III). This further *highlights the critical role of addressing the mismatch problem in the PFL.*

## 4.4 Disentangling Model Capacity and Training Strategy Effects

FedPFT introduces an FTM, which slightly increases model capacity compared to standard FL backbones. A natural concern is whether the performance gains arise simply from this added capacity, or if they can be achieved by applying existing fine-tuning strategies to the same architecture. To disentangle these factors, we reuse the architecture of FedPFT but train it with different strategies: (i) standard FedAvg without personalization

Table 4: Test accuracy (%) of FedAvg variants using the FedPFT backbone under Dirichlet non-IID setting on CIFAR-100.

| Methods | $\alpha = 0.1$ | $\alpha = 0.5$ | $\alpha = 1.0$ |
|---|---|---|---|
| FedPFT-Avg | 32.66±0.84 | 31.20±1.20 | 32.69±0.77 |
| FedPFT-Avg-Full FT | 57.90±0.52 | 38.44±1.26 | 36.10±0.08 |
| FedPFT-Avg-Prompt FT | 56.76±0.46 | 38.88±1.00 | 36.62±0.18 |

ization (**FedPFT-Avg**), (ii) full local fine-tuning after FedAvg (**FedPFT-Avg-Full FT**), and (iii) prompt-only fine-tuning after FedAvg (**FedPFT-Avg-Prompt FT**). Table 4 reports the results on CIFAR-100. Compared with FedAvg and FedPFT in Table 1, none of these alternatives match the performance of FedPFT, despite using the same model capacity.

These results demonstrate two key insights: (1) *Model capacity alone does not explain the improvements of FedPFT*, in fact, larger models are harder to train under limited local data. (2) *Simple*

*fine-tuning strategies, including prompt tuning, fail to achieve competitive results*, highlighting the necessity of our training-time alignment design.

## 4.5 Impact of Prompts on Feature Representation

In this section, we analyze how the introduced personalized prompts influence feature representation.

**The impact on the alignment of features with downstream tasks.** We visualize the features transformed by different prompts in FedPFT+Con using t-SNE. The experimental setup is consistent with Appendix I. The results are shown in Fig. 5. Larger markers in the figures represent feature centroids of corresponding classes for each client.

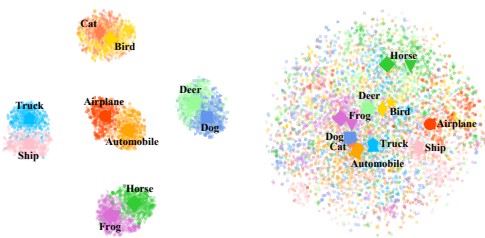

It is evident that the features transformed by different prompts adapt to the specific requirements of their respective downstream tasks: features from classification prompts $p_\kappa$ align more with client data distribution than image similarity; for instance, two classes within a client may appear close even if the images are not visually similar. Additionally, these features exhibit well-defined decision boundaries between classes, as expected in classification tasks. In contrast, features from contrastive learning prompts $p_\rho$ are more aligned with image similarity. For example, in Fig. 5(b), the feature centroids of 'cat' and 'dog' are close, while those of 'airplane' and 'dog' are farther apart, consistent with contrastive learning principles.

(a) Features transformed by classification prompts $p_\kappa$

(b) Features transformed by contrastive learning prompts $p_\rho$

Figure 5: The effect of different prompts on feature space.

**The impact on the linear separability of features.** We assess the linear separability of features at three stages during the forward propagation process: 'None' represents the features extracted by $\phi$. '$p_\kappa$' represents the features transformed by the FTM using classification prompts. '$p_\rho$' represents the features transformed by the FTM using contrastive learning prompts. The results are detailed in Table 5. Interestingly, the accuracies across different prompt conditions are generally similar, suggesting that the prompts do not carry additional knowledge to enhance feature separability.

These experiments confirm that *prompts work by transforming features into the required format to align with downstream tasks*, rather than improving feature separability. This finding is consistent with the motivation of our paper.

Table 5: The effect of prompts $p_\kappa$ and $p_\rho$ on linear probe accuracy.

|  | CIFAR-10 | | | CIFAR-100 | | |
| --- | --- | --- | --- | --- | --- | --- |
| Prompt | 0.1 | 0.5 | 1.0 | 0.1 | 0.5 | 1.0 |
| None | 87.69% | 77.12% | 73.93% | 64.08% | 46.50% | 40.79% |
| $p_\kappa$ | 87.83% | 77.25% | 74.02% | 64.12% | 46.43% | 40.95% |
| $p_\rho$ | 87.82% | 77.25% | 74.02% | 64.18% | 46.40% | 40.95% |

## 4.6 Experiments in Feature Shift Scenarios

We conduct experiments on two feature shift datasets, PACS [17] and DomainNet [38]. PACS and DomainNet have four and six domains, respectively, with each domain assigned to one client. The number of clients corresponds to the number of domains. For each client, we allocate 1000 training samples and 500 testing samples. The experimental results are shown in Table 6.

**Existence of the mismatch phenomenon.** The results show that in both datasets, there are still noticeable gaps between Origin Acc. and Match Acc. across all methods, especially in FedAvg. This indicates that the mismatch problem persists in feature shift scenarios and is a major reason for the suboptimal performance of FedAvg.

**Superiority of our method.** From the perspective of mismatch degree, the small gap between Origin Acc. and Match Acc. in FedPFT demonstrates its effectiveness in addressing the mismatch problem in feature shift non-IID scenarios. In terms

Table 6: Comparison of different methods in feature shift non-IID scenarios.

| | PACS | | DomainNet | |
| --- | --- | --- | --- | --- |
| Methods | Origin Acc. | Match Acc. | Origin Acc. | Match Acc. |
| FedAvg | 71.48% | 75.49% | 63.53% | 67.17% |
| FedPer | 74.86% | 75.31% | 65.70% | 65.67% |
| FedBN | 73.91% | 74.71% | 67.57% | 68.57% |
| FedCAC | 74.94% | 75.94% | 67.80% | 68.53% |
| FedPFT | 77.67% | 77.64% | 70.37% | 70.57% |

of Origin Acc., FedPFT significantly outperforms SOTA methods (*e.g.*, by up to 2.73% on PACS), further highlighting the superiority of FedPFT in addressing the feature shift non-IID problem.

### 4.7 Robustness

In this section, we evaluate the robustness of FedPFT under partial client participation. We conduct experiments on CIFAR-10, CIFAR-100, and Tiny ImageNet with $\alpha = 0.5$, considering scenarios where only a random 50%, 70%, and 90% of clients participate in training each round. The experimental results are presented in Table 7.

Table 7: Accuracy (%) of FedPFT with partial client participation. Values in '()' indicate the performance change relative to 100% client participation, as presented in Table 1.

| Datasets | 90% | 70% | 50% |
|---|---|---|---|
| CIFAR-10 | 73.88±1.84 (-0.22) | 74.21±1.45 (+0.11) | 74.33±1.38 (+0.23) |
| CIFAR-100 | 45.74±0.32 (+0.87) | 45.46±1.14 (+0.59) | 45.87±0.81 (+1.00) |
| Tiny | 28.53±0.62 (-0.08) | 29.24±0.16 (+0.63) | 29.90±0.10 (+1.29) |

The results indicate that FedPFT maintains high accuracy even when only a subset of clients participate in training. Notably, on CIFAR-100 and Tiny ImageNet, performance may even improve under reduced client participation. This is likely because fewer participating clients per round can mitigate the impact of non-IID data distribution on the global model. These findings demonstrate the robustness of FedPFT in scenarios with partial client participation. For a detailed comparison with SOTA methods under different participation ratios, please refer to Appendix N.

### 4.8 Training Efficiency

As illustrated in Table 8, FedAvg uploads the feature extractor $\phi_i$ and classifier $h_{\kappa,i}$ in each round. FedPFT adds FTM $\tau_i$, increasing communication overhead by 20.47% for ResNet-8 and 21.17% for ResNet-10. While FedPFT incurs additional communication costs, it is important to weigh it against the performance enhancements and flexibility offered by $\tau_i$. The improved accuracy and robustness

Table 8: The communication cost of each client in FedAvg and FedPFT in one round. The percentages in '()' represent the increase compared to FedAvg.

| Models | $\phi_i$ | $\tau_i$ | $h_{\kappa,i}$ | FedAvg | FedPFT |
|---|---|---|---|---|---|
| ResNet-8 | 1.24M | 0.26M | 25.70K | 1.27M | 1.53M (20.47%) |
| ResNet-10 | 4.91M | 1.05M | 51.30K | 4.96M | 6.01M (21.17%) |

to non-IID data might justify the additional costs in scenarios where model performance is critical.

We also evaluate computation efficiency. FedPFT incurs slightly higher per-round computation time than FedAvg, but remains computationally more efficient than most SOTA methods. Please refer to the Appendix P for details.

## 5 Conclusion

We identify that in non-IID scenarios, the fundamental cause of the global model's poor performance on client data is the mismatch between local features extracted by the feature extractor and the global classifier. This mismatch not only reduces model inference accuracy but also exacerbates inter-client gradient divergence during training, hindering model aggregation and ultimately rendering the feature extractor suboptimal for client data. To address this issue throughout training, we propose FedPFT, a novel PFL method that incorporates a prompt-driven feature transformation module. In each training round, FedPFT first trains the prompts to align local features with the classifier, followed by updating model parameters. Our experiments demonstrate that FedPFT not only effectively resolves the mismatch but also improves feature extractor quality, leading to significant performance gains over state-of-the-art methods.

## Acknowledgments

This work was supported by the National Natural Science Foundation of China under Grants 62372028 and 62372027.

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

# A Formal Definition of Feature–Classifier Mismatch

We formally define the feature–classifier mismatch as:

$$\text{Mismatch}(f, W) = \mathbb{E}_{(x,y) \sim D_i} \big[ \mathcal{L}(f', W) - \mathcal{L}(f, W) \big], \tag{7}$$

where $D_i$ denotes the data distribution of client $i$, $f$ represents the original local features extracted by the feature extractor, $W$ is the classifier, $f'$ is the transformed feature generated by the feature transformation module (*e.g.*, a linear transformation layer), and $\mathcal{L}(\cdot, \cdot)$ is the loss function (*e.g.*, cross-entropy).

This metric quantifies the improvement in performance (*i.e.*, reduction in classification loss) when local features are aligned with the classifier. Since loss values are often less intuitive, we report the accuracy gap between "Origin Acc." and "Match Acc." (defined in the caption of Figure 1) as a practical measure of mismatch throughout the paper. To complement this definition, we also explore additional metrics for quantifying feature–classifier mismatch, as discussed in Appendix B.

# B Further Discussion on Feature–Classifier Mismatch

Beyond accuracy-based measures, we also compute the cosine similarity between local features and their corresponding classifier weights. As shown in Table 9, FedPFT achieves values comparable to strong baselines such as FedBABU-FT.

Table 9: Cosine similarity between local features and their corresponding classifier weights in different methods on CIFAR-10.

| Scenarios | FedAvg | FedAvg-FT | FedBABU | FedBABU-FT | FedPFT |
|---|---|---|---|---|---|
| $\alpha = 0.5$ | 0.20 | 0.26 | 0.28 | 0.39 | 0.42 |
| $\alpha = 1.0$ | 0.22 | 0.24 | 0.31 | 0.37 | 0.36 |

However, this similarity metric is not strongly correlated with accuracy or loss, as it can be influenced by confounding factors such as feature norm and classifier bias terms. For this reason, we do not adopt it as the primary definition of mismatch in our paper. Instead, we define mismatch through the performance gap before and after feature transformation, which directly reflects model effectiveness. Our method is therefore designed to learn a transformation module that minimizes classification loss, rather than enforcing explicit angular constraints.

# C More Explanation of the Experiments Discussed in the Introduction

To facilitate understanding of the experiments mentioned in the introduction, we give a toy example to visualize the models used to calculate Origin Acc., Probe Acc., and Match Acc. in Figure 6.

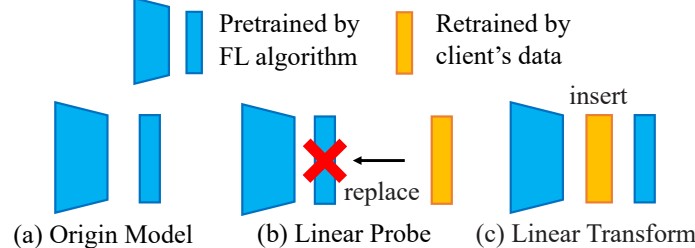

Figure 6: A toy example illustrating the model structures used to calculate Origin Acc., Probe Acc., and Match Acc.

Figure 6(a) represents the model trained using the FL algorithm, where the accuracy measured on the client's local data is referred to as Origin Acc. Figure 6(b) illustrates the model obtained from the linear probe experiment, where the classifier in the FL pre-trained model is replaced by a linear classifier retrained on the client data. The accuracy corresponding to this model is referred to as

Probe Acc. Figure 6(c) depicts the model obtained from the linear transform experiment, where a linear transformation layer, retrained on client data, is inserted between the feature extractor and classifier of the FL pre-trained model. The accuracy of this model is referred to as Match Acc.

## D  Pseudo-code of FedPFT

The pseudo-code of FedPFT is summarized in Algorithm 1.

---

**Algorithm 1** FedPFT

---

**Input:** Each client's initial personalized prompts $p_{\kappa,i}^{(0)}$; The initial global models $\{\phi^{(0)}, \tau^{(0)}, h_\kappa^{(0)}\}$; Client Number $N$; Total round $T$; Epochs of two learning phases $R_f$ and $R_a$.
**Output:** Personalized model $\{\phi^{(T)}, \tau^{(T)}, h_\kappa^{(T)}, p_{\kappa,i}^{(T)}\}$ for each client.
**for** $t = 0$ to $T - 1$ **do**
    **Client-side:**
    **for** $i = 1$ to $N$ **in parallel do**
        Initializing $\{\phi_i^{(t)}, \tau_i^{(t)}, h_{\kappa,i}^{(t)}\}$ with $\{\phi^{(t)}, \tau^{(t)}, h_\kappa^{(t)}\}$.
        Updating $\{\tau_i^{(t)}, p_{\kappa,i}^{(t)}\}$ by Eq.(4) for $R_f$ epochs to obtain $\{\tau_i^{(t')}, p_{\kappa,i}^{(t+1)}\}$.
        Updating $\{\phi_i^{(t)}, \tau_i^{(t')}, h_{\kappa,i}^{(t)}\}$ by Eq.(5) for $R_a$ epochs to obtain $\{\phi_i^{(t+1)}, \tau_i^{(t+1)}, h_{\kappa,i}^{(t+1)}\}$.
        Sending $\{\phi_i^{(t+1)}, \tau_i^{(t+1)}, h_{\kappa,i}^{(t+1)}\}$ to the server.
    **end for**
    **Server-side:**
    Aggregating a set of global model $\{\phi^{(t+1)}, \tau^{(t+1)}, h_\kappa^{(t+1)}\}$.
    Sending $\{\phi^{(t+1)}, \tau^{(t+1)}, h_\kappa^{(t+1)}\}$ to each client $i$.
**end for**

---

## E  Details of Combining FedPFT with Contrastive Learning

Following Section 3.6 of the main text, this section provides details of the combination of FedPFT with contrastive learning, including the definition of $L_{\text{Con}}$ and the optimization process of Eq. (6).

### E.1  Definition of $L_{\text{Con}}$

We adopt the Momentum Contrast (MoCo) framework [8] for contrastive learning. The associated contrastive loss function is defined as:

$$L_{\text{Con}}(\phi, \tau, p_{\rho,i}, h_\rho; x) = -\log \frac{\exp(q \cdot k_+ / \beta)}{\sum_{j=0}^{K} \exp(q \cdot k_j / \beta)}, \text{where } x \sim d_i. \tag{8}$$

In this formula, $h_\rho$ is the projection head used for contrastive learning. $q = h_\rho \circ \tau([\phi(x'), p_{\rho,i}])$ represents the query vector, and $k_+ = \tilde{h}_\rho \circ \tau([\tilde{\phi}(x''), p_{\rho,i}])$ denotes the positive key vector. Here, $x'$ and $x''$ are augmented versions of the sample $x$, $\tilde{\phi}$ and $\tilde{h}_\rho$ refer to the momentum-updated encoder and projection head, respectively. $\beta$ is a temperature hyperparameter, and $K$ is the number of negative samples drawn from MoCo's queue, comprising the set $\{k_j\}_{j=0}^{K}$.

### E.2  Optimization Process of FedPFT+Con

As discussed in Section 3.5 of the main text, FedPFT employs an alternating training strategy. In FedPFT+Con, we extend this approach by incorporating contrastive learning into the optimization process.

**Feature transformation phase.** In this phase, FedPFT+Con additionally utilizes $L_{\text{Con}}$ to update $\phi_i$, $\tau_i$ and $h_{\rho,i}$ to enhance the feature quality. The objective can be formulated as:

$$\min_{p_{\kappa,i}, \tau_i, \phi_i, h_{\rho,i}} L_{\text{CE}}(\tau_i, p_{\kappa,i}; \phi_i, h_\kappa, d_i) + L_{\text{Con}}(\phi_i, \tau_i, h_{\rho,i}; p_{\rho,i}, d_i). \tag{9}$$

**Model training phase.** FedPFT+Con additionally updates $p_{\rho,i}$ in this phase to align the features with the contrastive learning task, reducing interference from the classification task. Its training objective can be formulated as:

$$\min_{\phi_i,\tau_i,h_{\kappa,i},p_{\rho,i}} L_{\text{CE}}(\phi_i,\tau_i,h_{\kappa,i};p_{\kappa,i},d_i) + L_{\text{Con}}(p_{\rho,i},\tau_i;\phi_i,h_{\rho,i},d_i). \tag{10}$$

Figure 7 illustrates the training process of the contrastive learning and classification tasks in FedPFT+Con. The blue modules represent components from FedPFT, while the orange modules represent additional components introduced in FedPFT+Con. Solid arrows indicate forward propagation and dashed arrows represent backpropagation.

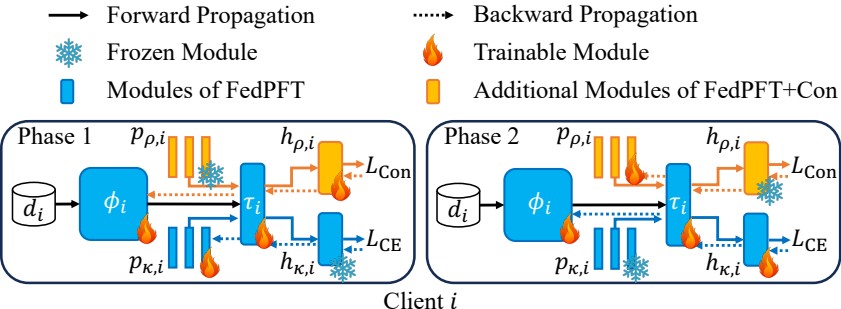

Figure 7: Training process of FedPFT+Con in each client $i$.

# F  Experiment Setup

## F.1  Introduction to non-IID Scenarios

**Pathological non-IID.** In this setting, each client is randomly assigned data from a subset of classes with equal data volume per class. For the CIFAR-10, CIFAR-100, and Tiny ImageNet datasets, we assign 2, 20, and 40 classes of data to each client, respectively.

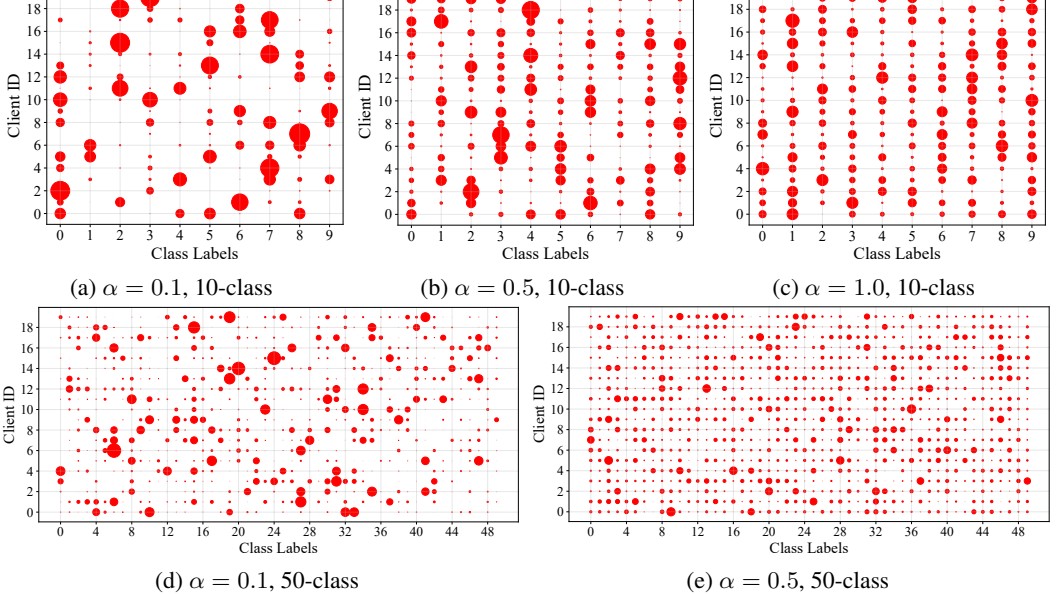

(a) $\alpha = 0.1$, 10-class    (b) $\alpha = 0.5$, 10-class    (c) $\alpha = 1.0$, 10-class

(d) $\alpha = 0.1$, 50-class    (e) $\alpha = 0.5$, 50-class

Figure 8: Visualization of data partitioning in Dirichlet non-IID scenarios with different $\alpha$.

**Dirichlet non-IID.** This is a commonly used setting in current FL research [51, 50, 41]. In this scenario, the data for each client is generated from a Dirichlet distribution denoted as $Dir(\alpha)$. As the value of $\alpha$ increases, the class imbalance within each client's dataset progressively decreases. This Dirichlet non-IID setting enables the evaluation of different methods across a broad spectrum of non-IID conditions, reflecting various degrees of data heterogeneity.

For a clearer, more intuitive understanding, we involve 20 clients with 10-class and 50-class datasets to visualize the data distribution among clients with varying $\alpha$ values. As depicted in Figure 8, the horizontal axis labels the data class indices, while the vertical axis lists the client IDs. Each red dot indicates the class data assigned to a client, with larger dots signifying a higher volume of data in that class.

### F.2 Introduction to Comparative Methods

FedBABU [37], FedETF [27], and FedFA [61] focus on enhancing the feature extractor of the global model. In FedBABU and FedETF, the classifier is frozen during the training of the global model. FedFA utilizes prototypes to align client feature extractors and calibrate the classifier. FedAMP [10] is a weighted-aggregation-based method where clients with similar data distributions are given higher aggregation weights during model aggregation. Because it mainly encourages the collaboration of clients with similar data distribution, it is a method that pays more attention to the local data distribution of clients. FedPer [1], FedRep [4], FedBN [26], FedRoD [3], and FedCAC [50] are parameter-decoupling-based methods, which personalize the global model by retaining certain parameters locally based on FedAvg. FedRoD additionally introduces a balanced global classifier to obtain assistance from other clients, alleviating the overfitting issue caused by personalized classifiers alone. pFedSD [12] and pFedGate [2] are fine-tuning-based methods that adapt the global model to local data through fine-tuning. pFedSD directly fine-tunes the global model by distilling local models, while pFedGate trains an additional gating network and applies it to the global model. pFedPT [18], a prompt-based method, can also be viewed as a fine-tuning approach, enhancing the global model's adaptation to local data distributions by adding prompts to images. FedAvg-FT, FedBABU-FT, and FedFA-FT are methods where all clients collaboratively train a global model during the first stage, followed by client-specific classifier fine-tuning of the global model to obtain personalized models. FedETF-Per uses the official fine-tuning method to obtain personalized models.

### F.3 Hyperparameter Settings in Different Methods

For the unique hyperparameters of each baseline method, we utilize the optimal parameter combinations reported in their respective papers. For learning rates, we adjust within {1e-1, 1e-2, 1e-3}.

In FedPFT, to simplify the hyperparameter tuning process and enhance usability, we provide a default set of hyperparameters: for all scenarios, we set $n_\kappa = 10$ and $(R_f, R_a) = (4, 1)$. We use the SGD optimizer with a learning rate of 0.05 for the FTM and 0.1 for other components. In FedPFT+Con, for the Dirichlet non-IID scenario with $\alpha = 0.1$, we set $(R_f, R_a) = (3, 2)$, while in other scenarios, we use $(R_f, R_a) = (4, 1)$. The learning rate for the FTM remains 0.01, with other hyperparameters consistent with FedPFT. Unless otherwise specified, the above hyperparameter settings are used in our experiments, though fine-tuning these parameters for specific scenarios may yield better performance.

### F.4 Compute Resources

Experiments are implemented using PyTorch and conducted on 4x NVIDIA RTX 2080 GPUs. For the methods we compared, as well as FedPFT, a single training session requires 12-36 hours. For FedPFT+Con, the training process takes longer due to the use of the MoCo algorithm, which requires data augmentation that can only be executed on the CPU. Consequently, a single training session for FedPFT requires 48-72 hours.

## G Comparison with State-of-the-art Methods

We present the comparative results of FedPFT and FedPFT+Con against established methods on CIFAR-10, CIFAR-100, and Tiny ImageNet datasets under Pathological non-IID scenarios in Table 10.

Table 10: Test accuracy (%) of different methods under Pathological non-IID setting on CIFAR-10, CIFAR-100, and Tiny ImageNet.

| Methods | CIFAR-10 | CIFAR-100 | Tiny ImageNet |
|---|---|---|---|
| FedAvg | $54.33 \pm 3.03$ | $34.27 \pm 0.44$ | $18.05 \pm 0.23$ |
| Local | $85.85 \pm 0.93$ | $38.40 \pm 0.69$ | $16.20 \pm 0.30$ |
| FedAMP | $88.88 \pm 0.83$ | $38.36 \pm 0.79$ | $16.13 \pm 0.55$ |
| FedPer | $87.51 \pm 0.95$ | $41.54 \pm 0.74$ | $20.25 \pm 0.65$ |
| FedRep | $87.10 \pm 0.91$ | $40.63 \pm 0.74$ | $19.24 \pm 0.33$ |
| FedBN | $87.02 \pm 1.41$ | $47.75 \pm 1.03$ | $24.91 \pm 0.48$ |
| FedRoD | $88.06 \pm 1.70$ | $52.55 \pm 0.92$ | $32.25 \pm 0.80$ |
| pFedSD | $89.97 \pm 1.45$ | $52.30 \pm 1.18$ | $30.27 \pm 0.78$ |
| pFedGate | $89.15 \pm 0.76$ | $43.73 \pm 0.14$ | $22.42 \pm 0.83$ |
| FedCAC | $89.77 \pm 1.14$ | $49.07 \pm 0.87$ | $30.83 \pm 0.42$ |
| pFedPT | $86.29 \pm 1.11$ | $39.92 \pm 0.33$ | $21.38 \pm 0.98$ |
| FedPFT | $89.67 \pm 1.96$ | $\mathbf{57.62 \pm 1.18}$ | $\mathbf{36.13 \pm 1.32}$ |
| FedPFT+Con | $\mathbf{90.55 \pm 1.35}$ | $\mathbf{58.14 \pm 0.71}$ | $\mathbf{37.59 \pm 0.39}$ |

**Results in Pathological non-IID scenario.** This is an extreme setting where each client has data from only a subset of classes. This scenario is particularly pronounced in the CIFAR-10 dataset, where each client essentially performs a simple binary classification task. Here, clients can achieve decent performance by solely focusing on their local tasks ('Local'), even without collaboration with other clients. As such, methods that prioritize local data distribution, such as FedAMP, pFedSD, and pFedGate, perform well. In contrast, on CIFAR-100 and Tiny ImageNet datasets, as clients have more local classes with fewer samples per class, local tasks become more challenging. Effective collaboration with other clients becomes crucial. Consequently, methods such as FedRoD, which emphasize client collaboration, exhibit increasingly significant performance. FedAMP and pFedGate show considerable performance degradation. FedPer, FedRep, FedBN, and FedCAC, by personalizing certain parameters of FedAvg, enhance local performance by indirectly aligning local features with classifiers to some extent. However, as they do not address the mismatch issue, they compromise the performance of feature extractors to some extent, thereby limiting their performance to a moderate level across the three datasets. FedPFT aligns local features with the global feature space using classification prompts, enhancing both local feature-classifier alignment and inter-client collaboration effectiveness. It achieves competitive performance on CIFAR-10 and surpasses existing SOTA methods on CIFAR-100 and Tiny ImageNet. FedPFT+Con further incorporates contrastive learning tasks to enhance feature extractor performance, outperforming SOTA methods significantly across all datasets.

# H   Addressing Mismatch by Inserting Linear Layer

As discussed in Section 1 of the main text, a straightforward approach to address the mismatch problem is to insert a personalized linear transformation layer between the global feature extractor and the global classifier (FedAvg+Linear). In this section, we validate this method through experiments, with the results shown in Table 11.

By combining the results in Table 1 of the main text, we observe that FedAvg+Linear outperforms most SOTA methods on the CIFAR-10 dataset, demonstrating the effectiveness of addressing the mismatch problem during training. However, on the more challenging CIFAR-100 dataset, FedAvg+Linear underperforms several SOTA methods. This illustrates that a simple linear transformation is insufficient for complex datasets. Notably, on CIFAR-100 with $\alpha = 1.0$, FedAvg+Linear even underperforms FedAvg, highlighting that FedAvg+Linear tends to overfit the limited local training data due to the large number of personalized parameters introduced.

In comparison, FedPFT demonstrates superior performance across all scenarios. Leveraging the flexibility of the FTM, FedPFT+Con further enhances model performance, significantly outperforming FedAvg+Linear.

Table 11: Test accuracy (%) of FedAvg+Linear under Dirichlet non-IID on CIFAR-10 and CIFAR-100.

| | CIFAR-10 | | | CIFAR-100 | | |
|---|---|---|---|---|---|---|
| Methods | $\alpha = 0.1$ | $\alpha = 0.5$ | $\alpha = 1.0$ | $\alpha = 0.1$ | $\alpha = 0.5$ | $\alpha = 1.0$ |
| FedAvg | 60.39±1.46 | 60.41±1.36 | 60.91±0.72 | 34.91±0.86 | 32.78±0.23 | 33.94±0.39 |
| Local | 81.91±3.09 | 60.15±0.86 | 52.24±0.41 | 47.61±0.96 | 22.65±0.51 | 18.76±0.63 |
| FedAvg+Linear | 85.96±2.23 | 71.17±1.28 | 67.63±0.83 | 58.07±0.41 | 37.09±0.85 | 31.23±0.24 |
| FedPFT | **87.23±2.69** | **74.10±1.95** | **69.23±0.76** | **60.98±0.39** | **44.87±0.76** | **41.83±0.37** |
| FedPFT+Con | **88.60±2.19** | **77.54±1.88** | **74.81±0.77** | **62.03±1.41** | **47.98±0.78** | **44.29±0.74** |

# I Learned Features of Different Methods

In this section, we visually compare the feature quality extracted by different methods. We conduct experiments on the CIFAR-10 dataset with 10 clients and the data distribution is illustrated in Fig. 9(a). For each method, we visualize the feature vectors of testing data from different clients using t-SNE [47]. The results are shown in Fig. 9(b)-(f).

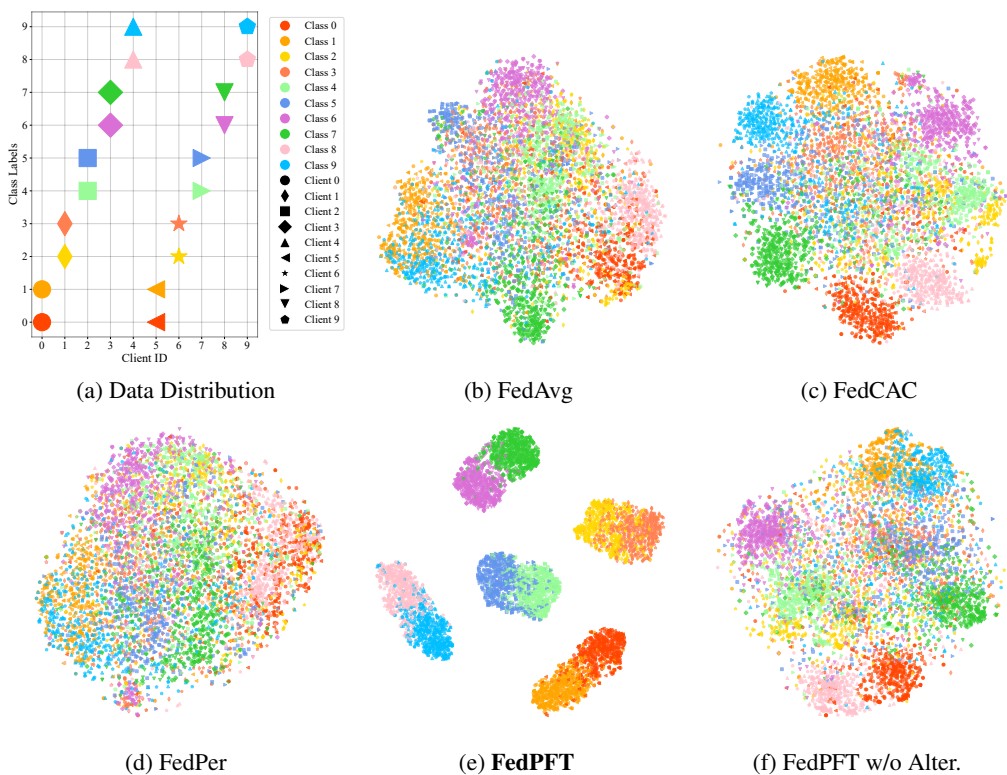

(a) Data Distribution    (b) FedAvg    (c) FedCAC

(d) FedPer    (e) **FedPFT**    (f) FedPFT w/o Alter.

Figure 9: t-SNE visualization of features extracted by different methods on the CIFAR-10 dataset. Different colors indicate various data categories, while distinct markers represent different clients.

FedAvg and FedCAC exhibit noticeable cluster structures of features but lack clear discriminative boundaries. FedPer shows overlapping features across various classes.

FedPFT exhibits clearer discriminative boundaries, which can be attributed to the alignment of local features with the global classifier achieved during local training. We also observe that data from the same class across different clients are mapped to the same positions in the feature space, indicating that our approach *not only resolves the mismatch issue but also addresses client feature discrepancies, as emphasized in many prior works* [61, 27, 54]. 'FedPFT w/o Alter.' denotes the variant without alternating training. Although it shows improved clustering compared to FedAvg,

its boundary discriminative quality is weaker than that of FedPFT, as it does not fully resolve the mismatch problem.

## J  Feature Separability of Different Methods

In this section, we delve deeper into the linear separability of features extracted by various PFL methods. We conduct linear probing experiments on the CIFAR-10 and CIFAR-100 to assess this property, with results detailed in Table 12.

Table 12: Linear probe accuracy (%) of different methods.

| Methods | CIFAR-10 | | | CIFAR-100 | | |
|---|---|---|---|---|---|---|
| | 0.1 | 0.5 | 1.0 | 0.1 | 0.5 | 1.0 |
| FedAvg | 85.01% | 72.52% | 68.38% | 59.50% | 37.40% | 32.33% |
| FedPer | 84.44% | 71.07% | 66.51% | 52.09% | 26.61% | 20.51% |
| FedBN | 84.52% | 70.15% | 66.51% | 57.86% | 35.24% | 30.28% |
| FedCAC | 85.22% | 71.56% | 66.98% | 56.86% | 34.64% | 29.35% |
| FedRoD | 82.79% | 67.07% | 63.12% | 56.88% | 33.99% | 29.22% |
| pFedSD | **85.86%** | 72.42% | 68.12% | 60.07% | 37.33% | 31.99% |
| FedPFT | 85.52% | **72.59%** | **69.57%** | **61.60%** | **43.14%** | **38.47%** |

It can be observed that the feature linear separability of most PFL methods is inferior to FedAvg. This suggests that although they partially alleviate the mismatch issue and achieve better model performance, the quality of the feature extractor is inevitably compromised due to their design, thereby limiting the full potential of PFL.

In stark contrast, FedPFT significantly improves the linear separability of features compared to FedAvg. It accomplishes this by fundamentally addressing the mismatch issue during the training process, thereby reducing model divergence among clients and the mutual interference between the feature extractor and classifier caused by misalignment.

## K  Comparison of FedPFT+Con with Two-stage Approach

In FedPFT+Con, we propose an FTM to coordinate the joint training of contrastive learning and classification tasks. To illustrate the superiority of this design, we introduce a baseline called 'Two-stage,' similar to [48], where contrastive learning training is conducted first, followed by classification task training after convergence. For fairness, in the two-stage method, we first perform 1000 rounds of contrastive learning training, followed by 1000 rounds of classification task training. The experimental results are depicted in Figure 10.

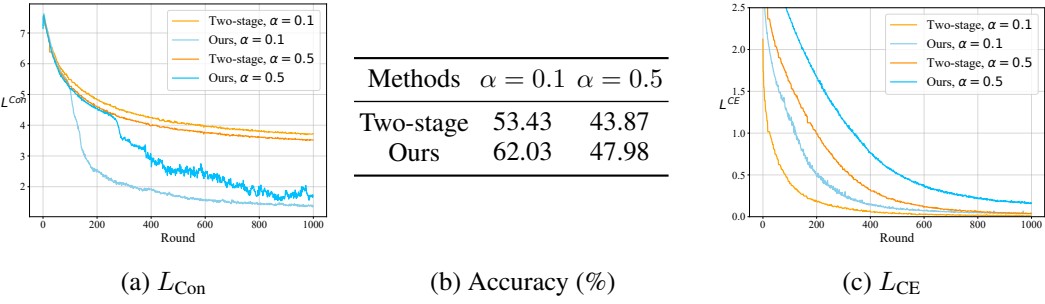

(a) $L_{\text{Con}}$          (b) Accuracy (%)          (c) $L_{\text{CE}}$

| Methods | $\alpha = 0.1$ | $\alpha = 0.5$ |
|---|---|---|
| Two-stage | 53.43 | 43.87 |
| Ours | 62.03 | 47.98 |

Figure 10: Comparison with two-stage approach on training $L_{\text{Con}}$, $L_{\text{CE}}$, and testing accuracy.

Firstly, from the perspective of the contrastive learning loss ($L_{\text{Con}}$), FedPFT+Con registers lower loss values compared to the Two-stage approach, suggesting that simultaneous training with the classification task enhances the efficacy of contrastive learning. Secondly, considering both Figure 10(b)

and Figure 10(c), our method exhibits significantly higher accuracy compared to the Two-stage approach. However, $L_{\text{CE}}$ converges to a higher training loss value, suggesting that in our design, contrastive learning tasks can alleviate overfitting issues in the classification task during training. These experiments demonstrate that our proposed approach can effectively coordinate both tasks, allowing them to assist each other. Importantly, these experiments also indicate that the significant performance improvement brought by contrastive learning in our method is largely attributed to the design of our FTM and training approach.

## L  Attention Weight Visualization

In the FTM of FedPFT and FedPFT+Con, self-attention mechanisms are employed to facilitate the integration of prompts with sample features. This section visualizes the attention weights to reveal how prompts influence the transformation process. We analyze 20 test samples from a single client on the CIFAR-10 dataset, with results depicted in Figure 11. Each row in the figure corresponds to the attention weights for the output feature $f'$ of a single sample. Columns represent the input dimensions of the FTM: the first column corresponds to the original input feature $f$, while subsequent columns relate to different prompts from the sets $p_{\kappa,i}$ or $p_{\rho,i}$.

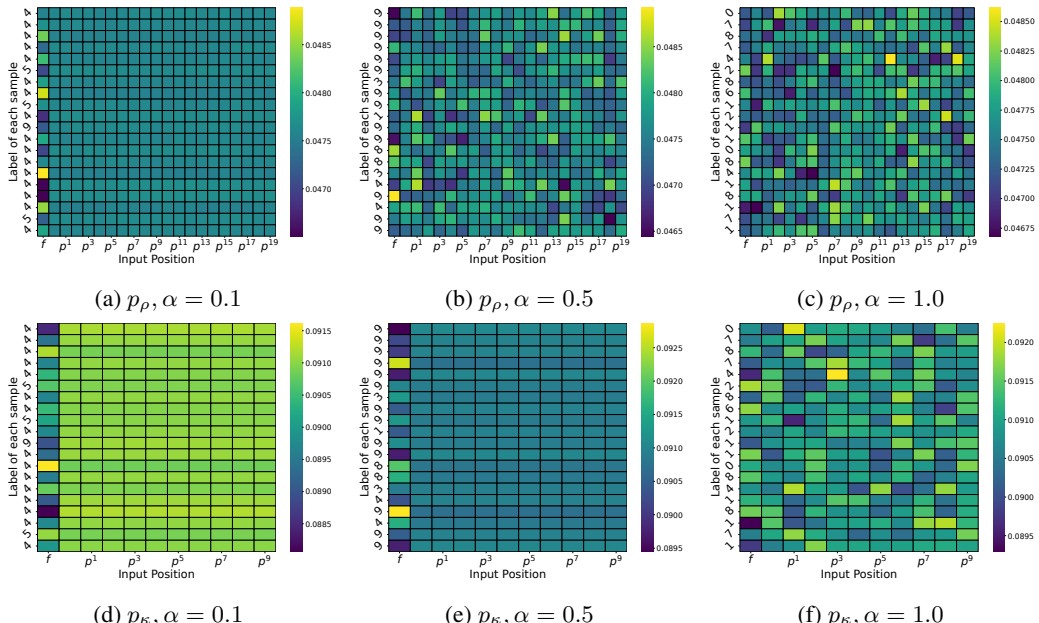

Figure 11: Visualize attention weights for different prompts in a client in the CIFAR-10 dataset under the Dirichlet non-IID scenario.

It can be observed that when $\alpha = 0.1$, indicating severe local class imbalances, each client has data from only a few classes. In this case, the feature transformation task is relatively simple, and the influence of different prompts on a sample is similar. As $\alpha$ increases, indicating more complex local tasks, the influence of prompts becomes more intricate. Particularly at $\alpha = 1.0$, it can be seen that each sample is affected differently by different prompts. This also indicates that our approach performs sample-level feature transformation.

## M  Expanding to more clients and More Complex Models

To demonstrate the scalability of FedPFT, we further conduct experiments on CIFAR-10 in Dirichlet non-IID scenarios with more clients and more complex models.

**Scaling to more clients.** We perform experiments with 100 clients, and the test accuracies of various methods are presented in Table 13.

Table 13: Accuracy (%) of different methods with 100 clients.

| Scenarios | FedPer | FedBN | FedRoD | FedCAC | FedPFT |
|---|---|---|---|---|---|
| $\alpha = 0.1$ | 84.68 | 85.51 | 87.58 | 87.40 | **88.43** |
| $\alpha = 0.5$ | 71.40 | 70.85 | 75.23 | 72.82 | **76.49** |
| $\alpha = 1.0$ | 66.92 | 67.18 | 70.99 | 69.06 | **72.53** |

**Scaling to more complex models.** We conduct experiments using ResNet-18, with the test accuracies of different methods shown in Table 14.

Table 14: Accuracy (%) of different methods with ResNet-18.

| Scenarios | FedPer | FedBN | FedRoD | FedCAC | FedPFT |
|---|---|---|---|---|---|
| $\alpha = 0.1$ | 82.80 | 78.10 | 83.50 | 82.60 | **83.52** |
| $\alpha = 0.5$ | 65.35 | 55.38 | 67.55 | 63.27 | **68.93** |
| $\alpha = 1.0$ | 61.83 | 56.88 | 62.05 | 60.53 | **64.95** |

In both experiments, FedPFT significantly outperforms state-of-the-art methods, highlighting the scalability of our approach.

## N    Partial Client Participation

As discussed in Section 4.7, we evaluated the robustness of FedPFT under partial client participation. In this section, we further consider a more challenging scenario where only 20% of clients participate in each round, and compare our method against state-of-the-art baselines. The results are reported in Table 15. As shown, FedPFT consistently outperforms all competing methods across different settings, demonstrating its strong robustness.

Table 15: Test accuracy (%) on CIFAR-100 with Dirichlet $\alpha = 0.5$ under different client participation ratios.

| Methods | 90% | 70% | 50% | 20% |
|---|---|---|---|---|
| FedAvg-FT | 41.50 | 40.88 | 41.95 | 40.23 |
| FedBABU-FT | 40.33 | 41.28 | 41.65 | 38.80 |
| FedRoD | 39.75 | 40.55 | 38.37 | 38.23 |
| FedPFT | 45.45 | 46.08 | 46.22 | 46.10 |

## O    Effect of Hyperparameters

In the previous experiments, we utilize the default hyperparameter combination. In this section, we verify how variations in these hyperparameters influence the performance of FedPFT.

### O.1    Effect of $n_\kappa$

$n_\kappa$ represent the number of prompts in $p_{\kappa,i}$ for each client. We examine the impact of this hyperparameter on the performance of FedPFT on CIFAR-10 and CIFAR-100 datasets. The experimental results are depicted in Figure 12.

FedPFT shows considerable robustness to variations in $n_\kappa$. On the CIFAR-10 dataset, changes in $n_\kappa$ have minimal impact on performance, suggesting that the model can effectively handle simpler data distributions even with fewer prompts. In contrast, on the more complex CIFAR-100 dataset, performance is initially limited by a small number of prompts, which may not sufficiently cover the diverse feature space required for effective feature transformation. As the number of

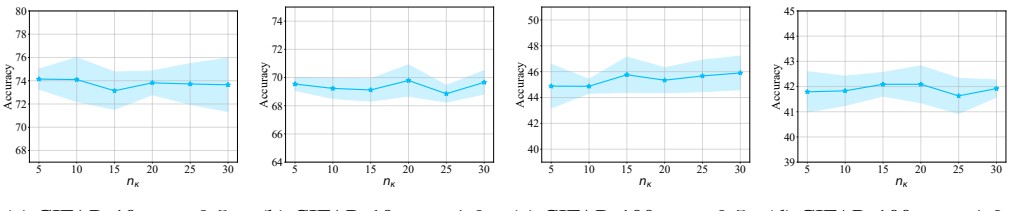

(a) CIFAR-10, $\alpha = 0.5$  (b) CIFAR-10, $\alpha = 1.0$  (c) CIFAR-100, $\alpha = 0.5$  (d) CIFAR-100, $\alpha = 1.0$

Figure 12: The effect of hyperparameter $n_\kappa$ on CIFAR-10 and CIFAR-100 in the Dirichlet non-IID scenario.

prompts increases, the model's ability to transform and adapt features improves, leading to enhanced performance.

## O.2  Effect of $R_f$ and $R_a$

$R_f$ and $R_a$ are used to control the number of training epochs for the two training stages. Since we set $R_f + R_a = R$, in this experiment, we only adjust $R_f$ to examine the impact of these two hyperparameters on model performance. The experimental results are illustrated in Figure 13.

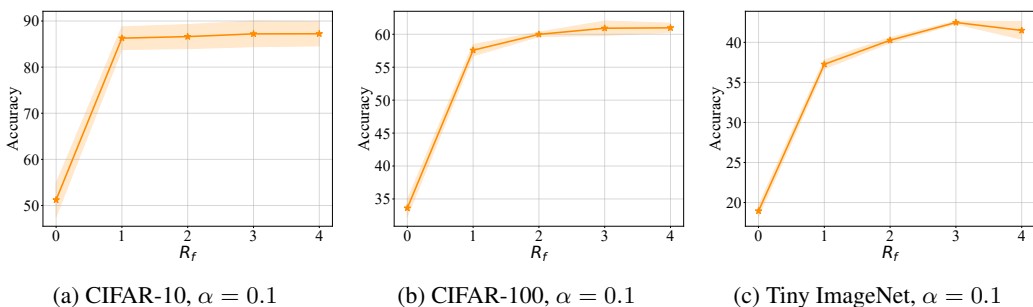

(a) CIFAR-10, $\alpha = 0.1$  (b) CIFAR-100, $\alpha = 0.1$  (c) Tiny ImageNet, $\alpha = 0.1$

Figure 13: The effect of hyperparameter $R_f$ on CIFAR-10, CIFAR-100, and Tiny ImageNet in the Dirichlet non-IID scenario with $\alpha = 0.1$.

When $R_f = 0$, it indicates that local features are not aligned with the global classifier before training the model parameters. Under this condition, the model performance is observed to be very poor. As $R_f$ gradually increases, the model performance initially improves but then declines in some scenarios, suggesting that $R_f$ balances the trade-off between the two training stages. When $R_f$ is too small, local features are not sufficiently transformed to match the classifier, resulting in the model being affected by the mismatch during the model training phase, which reduces the synergy between the feature extractor and classifier. On the other hand, when $R_f$ is too large, the model parameters are insufficiently trained, limiting the learning of local knowledge from clients.

In general, $R_f$ and $R_a$ are two hyperparameters that need careful adjustment, as they have a significant impact on the performance of FedPFT. Typically, in scenarios where clients' local tasks are simple, it may be appropriate to decrease the value of $R_f$. In other cases, we recommend using a larger $R_f$ value to fully align the local features with the global classifier.

## O.3  Impact of Local Epoch $E$

In this section, we extend our evaluation to explore the impact of increasing local update epochs ($E \in \{10, 15, 20\}$). The results are shown in the Table 16. As expected, larger $E$ values lead to performance degradation across all methods due to exacerbated client drift under non-IID settings. However, FedPFT exhibits smaller performance drops, highlighting its robustness.

This robustness stems from our two-phase training design: by resolving the feature–classifier mismatch before model updates, FedPFT ensures that training begins from an aligned feature space and

Table 16: Test accuracy (%) on CIFAR-10 with Dirichlet $\alpha = 1.0$ under different $E$.

| Methods | 10 | 15 | 20 |
|---|---|---|---|
| FedAvg-FT | 67.73 | 67.55 | 67.23 |
| FedBABU-FT | 68.30 | 68.13 | 67.85 |
| FedRoD | 67.23 | 67.05 | 66.80 |
| FedPFT | 69.43 | 69.32 | 69.10 |

mitigates interference between the feature extractor and classifier, thus reducing the adverse effects of drift.

## P    Computation Efficiency

We empirically evaluate the computational efficiency of our method on CIFAR-100 using ResNet-10, with results displayed in Table 17. We run each method for 100 rounds and calculate their average runtime per round. Each method exclusively utilizes a single machine during runtime. All experiments are conducted on four NVIDIA RTX 2080 GPUs.

Table 17: The average computation time per round for different methods on CIFAR-100.

| Methods | FedAvg | FedPer | FedBN | FedRoD | FedCAC | FedPFT |
|---|---|---|---|---|---|---|
| Time per round (s) | 50.10 | 51.81 | 52.65 | 54.76 | 55.35 | 51.61 |

FedPFT introduces the FTM, which adds some computational overhead. However, since it does not require updating the feature extractor during the feature learning phase, this helps reduce training costs to some extent. Overall, from the results in Table 17, FedPFT takes slightly longer to compute than FedAvg but is more time-efficient compared to SOTA methods.

## Q    Limitations and Future Work

In this paper, we primarily investigate PFL methods that derive personalized models based on a global model. We analyze the essential reasons these methods enhance performance from the perspective of mismatches between local features and classifiers. Although such methods occupy the mainstream in the current PFL field, it is necessary to admit that there are some PFL methods that are not based on global models, such as personalized-weight-aggregation-based methods, which are not explored in this study. Additionally, while this paper observes that personalizing a subset of parameters degrades the quality of the feature extractor, the underlying reasons for this phenomenon require further investigation.

## R    Theoretical Analysis

Since the main problem in Eq. (3.2) is non-convex, we focus on the factors affecting convergence in the non-convex setting.

### R.1    Problem Setup

Non-convex case analyses are provided, because our model is multi-layer transformer. Analyses are as follows.

We transform the problem into an unconditional bi-level optimization problem:

$$\min_{w} \mathbf{E}F(w) = \mathbf{E}_i\{F_i(w) := \min_{p_{\kappa,i}} \mathbf{E}_{d_i} L_{\text{CE}}(w, p_{\kappa,i}; d_i)\}$$

Table 18: The glossary of notations used in the theoretical analysis.

| Implication | Notation |
|---|---|
| Global / Local loss | $L$ / $L_i$ |
| Global / Local problem | $F$ / $F_i$ |
| Local Dataset on $i^{\text{th}}$ client | $\tilde{d}_i \in d_i$ |
| Feature extractor | $\phi$ |
| Feature transformation module | $\tau$ |
| Classification / Contrastive learning prompts | $p_\kappa$ / $p_\rho$ |
| Feature extractor & Feature transformation module & Classifier | $w$ |
| Classification / Contrastive learning task head | $h_\kappa$ / $h_\rho$ |
| Global / Local problem's gradient | $\nabla F(w)$ / $\nabla F_i(w)$ |
| Local gradient approximation | $g_{i,r}^t$ |
| Client number | $N$ |
| Local update epoch | $R$ |
| The number of clients sampled at each global epoch | $S$ |
| The set of clients sampled at global epoch $t$ | $\mathcal{S}_t$ |
| The actual learning rate of global problem | $\tilde{\eta}$ |
| The learning rate of local problem | $\eta$ |
| Approximated local gradient error's upper-bound | $\delta$ |
| Local-global gradient error's upper-bound | $\sigma_F$ |
| Index of client, local epoch and global epoch | $i \in [N], r \in [R], t \in [T]$ |

where $\mathbf{E}$ represents the expectation of all random variables, $\mathbf{E}_i$ means the expectation of client sampling, $\mathbf{E}_{d_i}$ is the local data sampling expectation, and we use $w = \{\phi, \tau, h_\kappa\}$ for simplification, based on the equivalence of block coordinate descent and gradient descent.

With contrastive learning the problem could be transformed into a similar problem with constrain. By Lagrange duality, the main problem is transformed as follows:

$$\min_{\phi,\tau,h_\kappa} \min_{\{p_{\kappa,i}\}_{i \in [N]}} \mathbf{E}_i \mathbf{E}_{d_i} L_{\text{CE}}(\phi, \tau, h_\kappa, p_{\kappa,i}; d_i)$$

$$\text{s.t. } \mathbf{E}_i \mathbf{E}_{d_i} L_{\text{Con}}(\phi, \tau; d_i) \leq H_{\text{Con}}$$

## R.2 Propositions

**Proposition R.1** ($L$-smooth). *If $f$ is $L$-smooth, $\forall x, y$ we have:*

$$\langle \nabla f(x) - \nabla f(y), x - y \rangle \leq L||x - y||^2$$
$$||\nabla f(x) - \nabla f(y)|| \leq L||x - y||$$
$$||\nabla f(x) - \nabla f(y)||^2 \leq 2L[f(x) - f(y)]$$
$$f(y) - f(x) - \langle \nabla f(x), y - x \rangle \leq \frac{L}{2}||y - x||^2$$

**Proposition R.2** (Jensen's inequality). *If $f$ is convex, we have the following inequality:*

$$\mathbf{E}_X f(X) \geq f(\mathbf{E}_X X).$$

*A variant of the general one shown above, given a group $\{x_i\}_{i \in [N]}$:*

$$|| \sum_{i \in [N]} x_i||^2 \leq N \sum_{i \in [N]} ||x_i||^2.$$

**Proposition R.3** (Triangle inequality). *The triangle inequality, where $|| \cdot ||$ is the norm, and $A, B$ is the elements in the corresponding norm space:*

$$||A + B|| \leq ||A|| + ||B||$$

**Proposition R.4** (Matrix norm compatibility). *The matrix norm compatibility,* $A \in \mathbf{R}^{a \times b}, B \in \mathbf{R}^{b \times c}, v \in \mathbf{R}^b$:

$$||AB||_m \leq ||A||_m ||B||_m$$
$$||Av||_m \leq ||A||_m ||v||$$

**Proposition R.5** (Peter Paul inequality). *$\forall x, y$ and $\forall \epsilon > 0$, we have the following inequality:*

$$2\langle x, y \rangle \leq \frac{1}{\epsilon}||x||^2 + \epsilon||y||^2$$

### R.3 Assumptions

**Assumption R.1** (L-smooth local objectives). *$\forall i, F_i$ is $L_F$-Smooth, the main proposition is shown in Prop. R.1. Notice that the $F_i$ is assumed to be L-smooth and non-convex, which matches the problem and neural network architecture setting in the main paper.*

**Assumption R.2** (Bounded local variance). *The local problem's gradient is assumed not to be too far from the global problem's gradient.*

$$\forall w, \mathbf{E}_i ||\nabla F_i(w) - \nabla F(w)|| \leq \sigma_F$$

**Assumption R.3** (Bounded approximated gradient). *The first-order approximation of the local problem's gradient $g_{i,r}^t$ should not be too far from the ground truth $\nabla F_i(w_{i,r}^t)$. In this assumption, the approximated error of the block coordinate descent in Algorithm 1 is bounded.*

$$\forall \{(i, r, t)\}, ||g_{i,r}^t - \nabla F_i(w_{i,r}^t)|| \leq \delta$$

### R.4 Lemmas

**Lemma R.1** (Bounded local approximation error). *If $\tilde{\eta} := \eta R \leq \frac{1}{2L_F}$, we have the following bound of client drift error:*

$$\frac{1}{NR} \sum_{i,r}^{N,R} \mathbf{E}||g_{i,r}^{(t)} - \nabla F_i(w^{(t)})||^2 \leq 2\delta^2 + 2^{R+3} L_F [3\tilde{\eta}^2 \sum_i^N \mathbf{E}||\nabla F_i(w^{(t)})||^2 + \frac{2\tilde{\eta}^2 \delta^2}{R}]$$

*Proof.* The client drift error on given $i^{\text{th}}$ client and its upper bound are as follows:

$$\begin{aligned}
&\mathbf{E}||g_{i,r}^{(t)} - \nabla F_i(w^{(t)})||^2 \\
\leq &2\mathbf{E}||g_{i,r}^{(t)} - \nabla F_i(w_{i,r}^{(t)})||^2 + 2\mathbf{E}||\nabla F_i(w^{(t)}) - \nabla F_i(w_{i,r}^{(t)})||^2 \quad (11) \\
\leq &2\delta^2 + 2L_F \mathbf{E}||w_{i,r}^{(t)} - w^{(t)}||^2
\end{aligned}$$

where the first inequality is by Proposition R.3 and the second one is by Assumption R.1.

For the last term in the upper bound, we have the iterative formulation as follows:

$$\begin{aligned}
&\mathbf{E}||w_{i,r}^{(t)} - w^{(t)}||^2 \\
=&\mathbf{E}||w_{i,r-1}^{(t)} - w^{(t)} - g_{i,r-1}^{(t)}||^2 \\
\leq&2\mathbf{E}||w_{i,r-1}^{(t)} - w^{(t)} - \eta\nabla F_i(w^{(t)})||^2 + 2\eta^2 \mathbf{E}||g_{i,r-1}^{(t)} - \nabla F_i(w^{(t)})||^2 \\
\leq&2(1 + \frac{1}{2R})\mathbf{E}||w_{i,r-1}^{(t)} - w^{(t)}||^2 + 2(1 + 2R)\eta^2 \mathbf{E}||\nabla F_i(w^{(t)})||^2 \\
&+ 4\eta^2 [\delta^2 + L_F^2 \mathbf{E}||w_{i,r}^{(t)} - w^{(t)}||^2] \\
=&2(1 + \frac{1}{2R} + 2\eta^2 L_F^2)\mathbf{E}||w_{i,r-1}^{(t)} - w^{(t)}||^2 + 4\eta^2 \delta^2 \\
&+ 2(1 + 2R)\eta^2 \mathbf{E}||\nabla F_i(w^{(t)})||^2
\end{aligned}$$

where the two inequalities are by Proposition R.3, Proposition R.5 and Eq. (11).

Take $\tilde{\eta} := \eta R \leq \frac{1}{2L_F}$, we recursively unroll the inequality as follows:

$$\mathbf{E}||w_{i,r}^{(t)} - w^{(t)}||^2$$

$$\leq 2(1 + \frac{1}{R})\mathbf{E}||w_{i,r-1}^{(t)} - w^{(t)}||^2 + 4\eta^2\delta^2 + 2(1 + 2R)\eta^2\mathbf{E}||\nabla F_i(w^{(t)})||^2$$

$$\leq [3\tilde{\eta}^2\mathbf{E}||\nabla F_i(w^{(t)})||^2 + \frac{2\tilde{\eta}^2\delta^2}{R}]2^{R+2}$$

where the inequality is unrolled and we use $\frac{1}{R} \leq 1$. Thus, we have:

$$\mathbf{E}||g_{i,r}^{(t)} - \nabla F_i(w^{(t)})||^2 \leq 2\delta^2 + 2^{R+4}\tilde{\eta}^2 L_F[3\sigma_F^2 + 3\mathbf{E}||\nabla F(w^{(t)})||^2 + \frac{\delta^2}{R}]$$

$$\square$$

## R.5 Theorem and Discussion

**Theorem R.2** (Non-convex and smooth convergence of FedPFT). *Let Assumption R.1, Assumption R.2 and Assumption R.3 hold, if $\tilde{\eta} := \eta R \leq \min\{\frac{1}{2L_F}, \hat{\eta}\}$ is taken, where $\hat{\eta} := \frac{N/S-1}{24(N-1)2^R}\sigma_F^2 - 1$, we have the following bound:*

$$\mathcal{O}(\mathbf{E}||\nabla F(w^{(\bar{t})})||^2) := \mathcal{O}(\frac{\Delta_F}{\hat{\eta}T} + \frac{2^{R/3}L_F^{1/3}(R\sigma_F^2 + \delta^2)^{1/3}\Delta_F^{2/3}}{T^{2/3}R^{1/3}} + (\frac{\sigma_F\sqrt{L_F(N/S-1)\Delta_F}}{\sqrt{TN}}) + \delta^2)$$

*Proof.*

$$\mathbf{E}F(w^{(t+1)}) - \mathbf{E}F(w^{(t)})$$

$$\leq \mathbf{E}\langle\nabla F(w^{(t)}), w^{(t+1)} - w^{(t)}\rangle + \frac{L_F}{2}\mathbf{E}||w^{(t+1)} - w^{(t)}||^2$$

$$= -\tilde{\eta}\mathbf{E}\langle\nabla F(w^{(t)}), g^{(t)}\rangle + \frac{\tilde{\eta}^2 L_F}{2}\mathbf{E}||g^{(t)}||^2$$

$$= -\tilde{\eta}\mathbf{E}||\nabla F(w^{(t)})||^2 - \tilde{\eta}\mathbf{E}\langle\nabla F(w^{(t)}), g^{(t)} - \nabla F(w^{(t)})\rangle + \frac{\tilde{\eta}^2 L_F}{2}\mathbf{E}||g^{(t)}||^2$$

$$\leq -\frac{\tilde{\eta}}{2}\mathbf{E}||\nabla F(w^{(t)})||^2 + \frac{\tilde{\eta}}{2}\mathbf{E}||\frac{1}{NR}\sum_{i,r}^{N,R}g_{i,r}^{(t)} - \nabla F_i(w^{(t)})||^2 + \frac{\tilde{\eta}^2 L_F}{2}\mathbf{E}||g^{(t)}||^2$$

$$\leq -\frac{\tilde{\eta}}{2}\mathbf{E}||\nabla F(w^{(t)})||^2 + \frac{\tilde{\eta}}{2}\mathbf{E}||\frac{1}{NR}\sum_{i,r}^{N,R}g_{i,r}^{(t)} - \nabla F_i(w^{(t)})||^2$$

$$+ \frac{3\tilde{\eta}^2 L_F}{2}\mathbf{E}[||g^{(t)} - \nabla F_i(w^{(t)})||^2 + ||\frac{1}{S}\sum_{i\in\mathcal{S}^{(t)}}\nabla F_i(w^{(t)}) - \nabla F(w^{(t)})||^2 + ||\nabla F(w^{(t)})||^2]$$

$$= -\frac{\tilde{\eta}(1 - 3\tilde{\eta}L_F)}{2}\mathbf{E}||\nabla F(w^{(t)})||^2 + \frac{\tilde{\eta}(1 + 3\tilde{\eta}L_F)}{2}\frac{1}{NR}\sum_{i,r}^{N,R}\mathbf{E}||g_{i,r}^{(t)} - \nabla F_i(w^{(t)})||^2$$

$$+ \frac{3\tilde{\eta}^2 L_F}{2}||\frac{1}{S}\sum_{i\in\mathcal{S}^{(t)}}\nabla F_i(w^{(t)}) - \nabla F(w^{(t)})||^2$$

$$\leq -\frac{\tilde{\eta}(1 - 3\tilde{\eta}L_F)}{2}\mathbf{E}||\nabla F(w^{(t)})||^2 + 3\tilde{\eta}^2 L_F\frac{N/S-1}{N-1}[\sigma_F^2 + ||\nabla F(w^{(t)})||^2]$$

$$+ \tilde{\eta}(1 + 3\tilde{\eta}L_F)[\delta^2 + 2^{R+3}\tilde{\eta}^2 L_F[3\sigma_F^2 + 3\mathbf{E}||\nabla F(w^{(t)})||^2 + \frac{\delta^2}{R}]]$$

where the four inequalities are respectively by $L_F$-smooth of $F := \mathbf{E}_i F_i$, Proposition R.5, Lemma R.1 and the similar classic Lemma 4 in [41].

Let $c_1 := 3\delta^2$, $c_2 := 3L_F\sigma_F^2\frac{N/S-1}{N-1}$, $c_3 := 2^{R+3}L_F[3\sigma_F^2 + \frac{\delta^2}{R}]$,

$$\mathbf{E}F(w^{(t+1)}) - \mathbf{E}F(w^{(t)}) \leq -\frac{\tilde{\eta}}{2}\{1 - [\frac{3}{2} - 3\frac{N/S-1}{N-1}\sigma_F^2 + 72 \times 2^R\tilde{\eta}]\}\mathbf{E}||\nabla F(w^{(t)})||^2$$
$$+ c_3\tilde{\eta}^3 + c_2\tilde{\eta}^2 + c_1\tilde{\eta}$$
$$\leq -\frac{\tilde{\eta}}{2}\mathbf{E}||\nabla F(w^{(t)})||^2 + c_3\tilde{\eta}^3 + c_2\tilde{\eta}^2 + c_1\tilde{\eta}$$

where let $\tilde{\eta} \leq \min\{\frac{1}{2L_F}, \hat{\eta}$, where $\hat{\eta} := \frac{2}{3\times 2^{R+4}}\frac{N/S-1}{N-1}\sigma_F^2 - 1\}$. Re-arranging the inequality above and accumulating, we have:

$$\frac{1}{2}\mathbf{E}||\nabla F(w^{(t)})||^2 \leq \mathbf{E}F(w^{(t+1)}) - \mathbf{E}F(w^{(t)}) + c_3\tilde{\eta}^2 + c_2\tilde{\eta} + c_1$$

$$\frac{1}{2T}\sum_{t=0}^{t=T-1}\mathbf{E}||\nabla F(w^{(t)})||^2 \leq \mathbf{E}F(w^{(T)}) - \mathbf{E}F(w^{(0)}) + c_3\tilde{\eta}^2 + c_2\tilde{\eta} + c_1$$

Let $\Delta_F = F(w^0) - F(w^*)$, where $w^*$ is the minimum of the main problem $\arg\min_w \mathbf{E}F(w)$. To measure the exact term of the bounds, we consider the following cases:

- $\frac{\Delta_F}{c_3T} \leq \tilde{\eta}^3$ or $\frac{\Delta_F}{c_2T} \leq \tilde{\eta}^2$, let $\tilde{\eta} = \min\{(\frac{\Delta_F}{c_3T})^{1/3}, (\frac{\Delta_F}{c_2T})^{1/2}\}$, we have:

$$\frac{1}{2}\mathbf{E}||\nabla F(w^{(t)})||^2 \leq \frac{c_3^{1/3}\Delta_F^{2/3}}{T^{2/3}} + (\frac{c_2\Delta_F}{T})^{1/2} + c_1$$

- $\frac{\Delta_F}{c_3T} \geq \tilde{\eta}^3$ and $\frac{\Delta_F}{c_2T} \geq \tilde{\eta}^2$, let $\tilde{\eta} = \hat{\eta}$, we have:

$$\frac{1}{2}\mathbf{E}||\nabla F(w^{(t)})||^2 \leq \frac{\Delta_F}{\hat{\eta}T} + \frac{c_3^{1/3}\Delta_F^{2/3}}{T^{2/3}} + (\frac{c_2\Delta_F}{T})^{1/2} + c_1$$

Uniformly sample a $\bar{t} \in [T] - 1$, we have the upper bound as follows:

$$\frac{1}{T}\sum_{t=0}^{T-1}\mathbf{E}||\nabla F(w^{(t)}||^2) = \mathcal{O}(\mathbf{E}||F(w^{(\bar{t})}||^2)$$

$$:= \mathcal{O}(\frac{\Delta_F}{\hat{\eta}T} + \frac{2^{R/3}L_F^{1/3}(R\sigma_F^2 + \delta^2)^{1/3}\Delta_F^{2/3}}{T^{2/3}R^{1/3}} + (\frac{\sigma_F\sqrt{L_F(N/S-1)\Delta_F}}{\sqrt{TN}}) + \delta^2)$$

$\square$

**Remark R.2.1.** *According to Theorem R.2, our proposed FedPFT converges at a sub-linear level. The linear term $\mathcal{O}(\frac{\Delta_F}{\hat{\eta}T})$ is affected by $\hat{\eta}$ and the initialization gap $\Delta_F$. The sub-linear term $\mathcal{O}(1/T^{2/3})$ is affected by $R$, especially when $R$ is large due to the exponential factor $2^R$. As the local approximation error of the gradient $\delta$ grows, both the convergence radius $\mathcal{O}(\delta)$ and the sub-linear term $\mathcal{O}(1/T^{2/3})$ are affected by the local optimizer selection significantly. Another sub-linear term $\mathcal{O}(\sqrt{T})$ is eliminated if $N/S - 1 = 0$ when all the clients are sampled. Otherwise, the sub-linear rate is mainly affected by $\sigma_F$.*

*FedPFT addresses the mismatch and aligns the training objectives across clients by introducing $p_{\kappa,i}$. Our design can effectively reduce differences in local gradients among clients during training, thereby reducing $\sigma_F$ and subsequently lowering the upper bound. During training, $p_{\kappa,i}$ incorporate information from the local datasets. By using them as part of the input, FedPFT effectively reduces the randomness in gradient computation, thereby lowering $\delta$ and consequently reducing the upper bound.*

## S  Empirical Support for Remark R.2.1.

To validate Remark R.2.1, we measure the standard deviation of local gradients within clients ($\delta$) and the gradient differences across clients ($\sigma_F$) in each training round. As shown in the Fig. 14, our proposed prompts significantly reduce both metrics, directly supporting the theoretical insights.

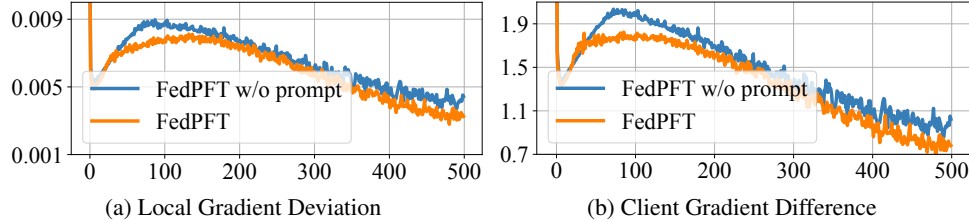

(a) Local Gradient Deviation         (b) Client Gradient Difference

Figure 14: Comparison of local gradient deviation and inter-client gradient difference with and without prompts in FedPFT on CIFAR-100.

