# OpenReview forum: "Tackling Feature-Classifier Mismatch in Federated Learning via Prompt-Driven Feature Transformation"
_NeurIPS.cc/2025/Conference — NeurIPS 2025 poster_

### Official Review · Reviewer_JoZC · 2025-06-13

**Clarity:** 2
**Significance:** 2
**Originality:** 2
**Rating:** 4
**Confidence:** 4

**Summary:**

This paper proposes FedPFT, a framework that uses personalized prompts to address the mismatch between local features and the global classifier during training in heterogeneous FL. FedPFT improves feature alignment and supports additional training objectives like contrastive learning, achieving up to 5.07% better performance and 7.08% further gains with collaborative contrastive learning.

**Questions:**

1) Why does a smaller parameter in the Dirichlet distribution lead to a more heterogeneous setting? In most previous papers, a smaller Dirichlet parameter typically implies a more heterogeneous data distribution.

**Ethical Concerns:**

["NO or VERY MINOR ethics concerns only"]

**Final Justification:**

The author solved most of my questions. And I found a typo and misundertanding in my proposed question. After discussed with the authors, the question is solved. Therefore I kept the positive rating.

**Quality:**

3

**Strengths And Weaknesses:**

Strengths:
1) The paper is well-motivated with clear and effective representations.
2) The experiments include extensive baselines for thorough evaluation.

Weaknesses:
1) The proposed method appears somewhat straightforward and lacks theoretical motivation and analysis.
2) The experiments are conducted on a relatively small scale, limiting the evaluation's generalizability.

Considering the efforts in Table 1, I tend to give a marginaly positive score.

Minor: There are also several works addressing feature and classifier misalignment, which are recommended to be included in the related works section for discussion. Some of these methods are designed for G-FL, while others are applicable to P-FL. To name a few:
* FedAWS:Federated learning with only positive labels
* FedMR: Federated learning under partially class-disjoint data via manifold reshaping
* FedRS: Fedrs: Federated learning with restricted softmax for label distribution non-iid data
* FedGELA: Federated learning with bilateral curation for partially class-disjoint data

---

> ### Author Rebuttal · Authors · 2025-07-31
>
> We sincerely thank the reviewer for valuable feedback. We try our best to carefully address each of the raised weaknesses and questions. Additionally, we would like to emphasize that the code for our method was submitted along with the original submission, and we commit to open-sourcing the additional experiments conducted during the rebuttal phase if this paper is accepted.
>
> ## Weakness 1: Lack of Theoretical Motivation and Analysis
>
> We thank the reviewer for the thoughtful comment. We acknowledge the importance of conducting thorough theoretical analysis to strengthen the foundation of our work. While our current motivation is primarily empirical, grounded in practical observations, we have included a preliminary theoretical convergence analysis in Appendix O of the original manuscript. We recognize that a more comprehensive theoretical study would further enhance the rigor of our approach, and we leave this as an important direction for future work.
>
> ## Weakness 2: Limited Scale of Experiments
>
> We appreciate the reviewer’s concern. We have provided **larger-scale experiments** in **Appendix K (Table 12 and Table 13)** of the original manuscript, including:
>
> - **100 participating clients**,
> - **Deeper model architectures**, such as **ResNet-18**
>
> For the reviewer’s convenience, we highlight these results here. As shown, FedPFT consistently outperforms SOTA baselines even under these more demanding configurations, demonstrating its robustness and scalability.
>
> | 100 clients  | FedPer | FedBN | FedRoD | FedCAC | FedPFT    |
> | ------------ | ------ | ----- | ------ | ------ | --------- |
> | $\alpha=0.1$ | 84.68  | 85.51 | 87.58  | 87.40  | **88.43** |
> | $\alpha=0.5$ | 71.40  | 70.85 | 75.23  | 72.82  | **76.49** |
> | $\alpha=1.0$ | 66.92  | 67.18 | 70.99  | 69.06  | **72.53** |
>
> | ResNet-18    | FedPer | FedBN | FedRoD | FedCAC | FedPFT    |
> | ------------ | ------ | ----- | ------ | ------ | --------- |
> | $\alpha=0.1$ | 82.80  | 78.10 | 83.50  | 82.60  | **83.52** |
> | $\alpha=0.5$ | 65.35  | 55.38 | 67.55  | 63.27  | **68.93** |
> | $\alpha=1.0$ | 61.83  | 56.88 | 62.05  | 60.53  | **64.95** |
>
> In the rebuttal phase, we further conducted larger-scale experiments with **200 clients**. As shown below, FedPFT continues to outperform SOTA methods under these settings as well.
>
> | 200 clients  | FedAvg-FT | FedBABU-FT | FedRoD | FedPFT |
> | ------------ | --------- | ---------- | ------ | ------ |
> | $\alpha=0.1$ | 86.20     | 87.04      | 85.68  | 87.90  |
> | $\alpha=0.5$ | 72.82     | 73.80      | 72.60  | 74.81  |
> | $\alpha=1.0$ | 70.04     | 71.31      | 69.80  | 72.32  |
>
> ## Minor: Related Works on Feature–Classifier Misalignment
>
> We thank the reviewer for pointing out these additional related works.
>
> We have carefully reviewed the references provided by the reviewer and found that these works primarily address **feature or classifier misalignment across clients**. In contrast, our method focuses on the **intra-client mismatch between local features and the global classifier**, which we believe is an orthogonal but equally important problem.
>
> We will include a dedicated discussion in the revised version comparing our approach with feature/classifier alignment methods, incorporating the references suggested by you and other reviewers.
>
> ## Question 1: Dirichlet Parameter and Data Heterogeneity
>
> We thank the reviewer for the question. In federated learning, a smaller Dirichlet parameter ($\alpha$) leads to sparser and more concentrated class distributions on each client—that is, clients are more likely to receive data from only a few classes, resulting in stronger inter-client heterogeneity. Conversely, larger $\alpha$ values yield more balanced and overlapping distributions across clients.
>
> We also note that more details about the Dirichlet-based non-IID setting are provided in Appendix D.1, and Figure 8 provides a visual illustration to help readers better understand how varying $\alpha$ influences the data distribution.

---

> > ### Comment · Reviewer_JoZC · 2025-08-03
> > **Reply to the rebuttal**
> >
> > Thanks for the reply, there is a typo in question part."Why does a larger parameter in the Dirichlet distribution lead to a more heterogeneous setting? In most previous papers, a smaller Dirichlet parameter typically implies a more heterogeneous data distribution." I think it may due to the setting and evaluation of PFL that test set is also heterogeneous according to its personal distribution.
> >
> > Thanks for the rebuttal. The authors solved most of my questions.

---

> > > ### Author Response · Authors · 2025-08-03
> > >
> > > We sincerely thank the reviewer for the clarification and for thoughtfully revisiting the question. We are glad to hear that most of your concerns have been addressed, and we truly appreciate your positive feedback and helpful suggestions throughout the review process.

---

### Official Review · Reviewer_3APB · 2025-07-02

**Clarity:** 3
**Significance:** 2
**Originality:** 2
**Rating:** 4
**Confidence:** 3

**Summary:**

The paper addresses a challenge in Federated Learning (FL) caused by data heterogeneity, which leads to suboptimal performance of global models across diverse client distributions.
They identify that the primary cause of suboptimal performance is the mismatch between local features and the global classifier, rather than the inadequacy of the feature extractor or classifier themselves.

To address this issue, they propose FedPFT, a new Personalized Federated Learning (PFL) framework that uses personalized prompts and a shared self-attention-based feature transformation module (FTM) to align local features with the global classifier during training. The paper demonstrates through experiments that FedPFT outperforms state-of-the-art methods by up to 5.07%, and when combined with contrastive learning, the improvement can reach up to 7.08%.

**Questions:**

- Could you formally define “feature–classifier mismatch” and clarify how it differs from discrepancies investigated in prior works [1,2,3,4,5,6,7]?

- Please report a quantitative metric—such as cosine similarity between local features, classifier weights, or average gradient-angle divergence—and show how FedPFT and existing baselines change that metric during training.

- The robustness exp in Section 4.7 should be  compared  with other baselilnes, and more sparse client (eg, 10%, 20%) should be considered for the realistic problem setting.

- Have you measured communication cost or accuracy when applying FedPFT to out-of-distribution (OOD) clients?

- It is highly recommended to follow the experimental settings of large benchmarks, such as those in [8, 9], to ensure a fair comparison. Could you describe the problem setting more detaily in comparison to these benchcmarks?

[8] https://github.com/KarhouTam/FL-bench

[9] https://github.com/alibaba/FederatedScope

**Ethical Concerns:**

["NO or VERY MINOR ethics concerns only"]

**Final Justification:**

Based on the discussion, I have decided to increase my score slightly.

**Limitations:**

The authors have not discussed limitations of the work in the conclusion. I think the method is purely theoretical and there is no potential negative societal impact.

**Quality:**

3

**Strengths And Weaknesses:**

### Strengths
1. The paper offers a fresh perspective on the feature–classifier mismatch problem in FL.

2. FedPFT introduces a concrete framework—personalized prompts plus a self-attention FTM—that explicitly targets this mismatch.

3. Extensive experiments on CIFAR-10, CIFAR-100, and Tiny ImageNet under both Dirichlet and Pathological non-IID settings demonstrate clear performance gains over existing methods.

### Weaknesses
1. Although the authors promote feature–classifier mismatch as a new explanation for performance degradation, it is unclear precisely what this mismatch is or why it constitutes a novel, critical problem. Prior studies have already examined gaps between global and personalized models in feature, classifier, or weight space side [1, 2, 3, 4, 5, 6, 7].

2. To claim that FedPFT solves the mismatch problem, the paper should provide more evidence—e.g., extend Figure 1 (Gradient Difference) to include other SOTA models, or introduce quantitative metrics beyond pre-/post-accuracy. Performance gains alone are insufficient supports.

3. Showing that FedPFT pairs well with contrastive learning is useful but not, by itself, a novel contribution; it is incremental.

4. The personalized prompts and FTM increase both model complexity and communication cost. Figure 1 and Table 1 suggest that simple post-training or fine-tuning (FT) yields nearly the same accuracy boost with far less overhead.


[1] FedBABU: Toward Enhanced Representation For Federated Image Classification, ICLR 2022

[2] No Fear of Heterogeneity: Classifier Calibration for Federated Learning with Non-IID Data, NeurIPS 2021

[3]  Guiding The Last Layer in Federated Learning with Pre-Trained Models, NeurIPS 2023

[4] Personalized Fedrated Learning with Ehroretical Guarantees: A Model-Agnostic Meta-L earning Approach, NeurIPS 2020

[5] Personalized Federated Learning using Hypernetworks, ICML 2021

[6] Federated Learning with Partial Model Personalization, PMLR 2022

[7] Personalized Federated Learning via Variational Bayesian Inferennce, PMLR 2022

---

> ### Author Rebuttal · Authors · 2025-07-31
>
> We sincerely thank the reviewer for valuable feedback. We try our best to carefully address each of the raised weaknesses and questions. Additionally, we would like to emphasize that the code for our method was submitted along with the original submission, and we commit to open-sourcing the additional experiments conducted during the rebuttal phase if this paper is accepted.
>
> ## Weakness 1 & Question 1: Clarifying the Definition and Novelty of Feature–Classifier Mismatch
>
> We thank the reviewer for this important question. Below we provide a formal definition of feature–classifier mismatch, and explain in detail how it differs from the discrepancies studied in prior works [1–7].
>
> ### (1) Formal Definition of Feature–Classifier Mismatch
>
> We define feature–classifier mismatch as:
> $$
> \text{Mismatch}(f, W) = \mathbb{E}_{(x,y)\sim D_i}[\mathcal{L}(f', W) - \mathcal{L}(f, W)]
> $$
> where:
>
> - $D_i$ is the data distribution of client $i$.
> - $f$ denotes the original local features extracted by the feature extractor,
> - $W$ is the classifier,
> - $f'$ is the transformed feature produced by feature transformation module (e.g., linear transformation layer),
> - $\mathcal{L}(\cdot, \cdot) $ is the loss function (e.g., cross-entropy).
>
> This quantity measures how much the performance (i.e., classification loss) improves when we align the local features with the classifier. Since raw loss values are often not intuitive, we use the accuracy gap between **Origin Acc.**  and **Match Acc.** (as we defined in the caption of Figure 1) to reflect the mismatch degree throughout the paper.
>
> ### (2) Key Difference from Prior Work
>
> We highlight the **fundamental difference** between our work and existing literature.
>
> **1. Objective: Inter-Client vs. Intra-Client Alignment**
>
> - Prior works on feature/classifier [1, 2, 3] and weight space alignment [4, 5, 6, 7] mainly focus on **inter-client consistency**, aiming to reduce **client drift** by aligning models, features, or classifiers **across clients**.
> - In contrast, **our method explicitly targets intra-client mismatch**—that is, the misalignment between a client’s own local features and the shared global classifier. This is a new problem that has not been studied in federated learning but is critical for stable training and accurate inference under heterogeneous data.
>
> **2. Our Advantage**
>
> **Existing methods fail to address intra-client mismatch, whereas our approach simultaneously resolves intra-client mismatch and enhances inter-client consistency to mitigate client drift.** This joint resolution is crucial, because even with successful inter-client alignment, feature–classifier mismatches can still persist within individual clients, leading to degraded performance (as illustrated on the left side of Figure 1).
>
> - Our method tackles this issue at its root by first resolving the mismatch before local updates. This enables training to begin from an aligned feature space and avoids interference between the feature extractor and classifier—reducing client drift (Figure 1, bottom right).
> - Furthermore, by transforming local features to match the shared semantic space defined by the global classifier, our method **implicitly promotes inter-client feature consistency**, as evidenced by class-wise clustering across clients (Appendix Figure 9(e)).
>
> We thank the reviewer once again for the insightful question. We will incorporate the above clarification, formal definition, and comparative discussion into the revised version of the paper to better highlight the novelty and significance of our work.
>
>
>
> ## Weakness 2 & Question 2: Quantitative Evidence for Mismatch
>
> We thank the reviewer for this insightful suggestion. In response, we conducted two additional experiments during the rebuttal phase to provide more **quantitative evidence** beyond accuracy:
>
> ### (1) Gradient Divergence
>
> We measured the gradient divergence of the feature extractor across clients throughout training, and reported the **minimum, maximum, and average** divergence observed over the entire training process. (Figures are omitted per rebuttal policy.)
>
> |            | Min  | Max  | Average |
> | ---------- | ---- | ---- | ------- |
> | FedAvg     | 0.93 | 3.38 | 1.62    |
> | FedBABU    | 0.85 | 2.81 | 1.45    |
> | FedRoD     | 0.83 | 3.20 | 1.43    |
> | **FedPFT** | 0.67 | 1.82 | 1.36    |
>
> The results show that **FedPFT achieves consistently lower gradient divergence** compared to other methods. This suggests that resolving the feature–classifier mismatch prior to model updates leads to more consistent optimization directions, thereby mitigating client drift caused by non-IID data.
>
>
>
> ### (2) Feature–Classifier Cosine Similarity
>
> We also computed the **cosine similarity between local features and their corresponding classifier weights**. As shown in the table below, FedPFT achieves comparable values to strong baselines like FedBABU-FT.
>
> | CIFAR 10  | FedAvg | FedAvg-FT | FedBABU | FedBABU-FT | FedPFT |
> | --------- | ------ | --------- | ------- | ---------- | ------ |
> | Dir (0.5) | 0.20   | 0.26      | 0.28    | 0.39       | 0.42   |
> | Dir (1.0) | 0.22   | 0.24      | 0.31    | 0.37       | 0.36   |
>
> However, we note that **this similarity metric is not strongly correlated with accuracy or loss**. It can be influenced by confounding factors such as feature norm and classifier bias terms. This is why we **do not adopt this metric to define mismatch** in our paper.
>
> Instead, we define mismatch based on the **performance gap before and after feature transformation**, which directly reflects model effectiveness. Our method is also designed accordingly—to **learn a transformation module that minimizes classification loss**, rather than enforcing explicit angular constraints.
>
> ## Weakness 3: On the Role of Contrastive Learning
>
> We appreciate the reviewer’s observation. We clarify that the integration of contrastive learning is **not presented as a standalone contribution**, but rather intended to illustrate the **flexibility and extensibility** of our framework. In many real-world applications, achieving high accuracy is critical. Our method’s modular prompt-based design naturally supports the integration of complementary techniques—such as contrastive learning—to further boost performance when necessary. This makes FedPFT practically valuable and well-suited for a wide range of deployment scenarios.
>
>
>
> ## Weakness 4: On Overhead
>
> We thank the reviewer for raising this important point. We agree that FedPFT introduces additional model complexity and communication overhead due to the inclusion of the feature transformation module. This results in approximately 20% more resource consumption compared to baseline methods.
>
> While such overhead is relatively minor in cross-silo settings where clients typically have sufficient computational and communication resources, we acknowledge that it may pose challenges in resource-constrained cross-device scenarios. We sincerely appreciate the reviewer’s attention to this aspect.
>
> As part of future work, we plan to investigate more lightweight feature transformation modules to further reduce the overhead while preserving the effectiveness of our approach.
>
> ## Question 3: On Robustness Comparison
>
> We have conducted additional experiments under a **20% client participation** setting and compared FedPFT with SOTA methods. The results show that FedPFT maintains strong robustness and consistently outperforms baselines, even under sparse participation.
>
> | CIFAR100, Dir 0.5 | 90%       | 70%       | 50%       | 20%       |
> | ----------------- | --------- | --------- | --------- | --------- |
> | FedAvg-FT         | 41.50     | 40.88     | 41.95     | 40.23     |
> | FedBABU-FT        | 40.33     | 41.28     | 41.65     | 38.80     |
> | FedRoD            | 39.75     | 40.55     | 38.37     | 38.23     |
> | **FedPFT**        | **45.45** | **46.08** | **46.22** | **46.10** |
>
> ## Question 4: Communication and Accuracy on OOD Clients
>
> We thank the reviewer for raising this important point. In our current work, our primary focus is on achieving strong performance in in-distribution (ID) settings, and we have not conducted evaluations on out-of-distribution (OOD) clients. We agree that evaluating performance on out-of-distribution (OOD) clients is an interesting and meaningful extension. We consider this an important direction and plan to leave a comprehensive OOD evaluation to future work.
>
> ## Question 5: Comparison with FL Benchmarks [8, 9]
>
> We thank the reviewer for this valuable suggestion. We have carefully reviewed both **FederatedScope [9]** and **FL-Bench [8]**, and provide the following clarification:
>
> - **FederatedScope** is a highly modular FL platform that emphasizes system-level efficiency and rapid deployment. However, it currently provides limited support for state-of-the-art personalization algorithms, which are the focus of our work. As our contribution is algorithmic in nature, we chose not to base our implementation on FederatedScope.
> - **FL-Bench**, on the other hand, is a well-designed research benchmark that includes many strong personalized FL methods, standardized datasets, model architectures, and non-IID settings. Our experimental setup is **highly aligned with FL-Bench**, including:
>   - Use of public datasets (CIFAR-10, CIFAR-100, TinyImageNet);
>   - Adoption of Dirichlet and feature-shift non-IID partitions;
>   - Use of standard model architectures (e.g., ResNet);
>   - Client counts of 40 and 100 for small- and large-scale evaluations;
>   - Including many baselines that are also part of FL-Bench, such as FedRep, FedPer, FedBN, FedRoD, and FedBABU.
>
> Although our experiments are not strictly based on the FL-Bench codebase, our design choices are largely consistent with its experimental protocols, enabling fair and meaningful comparisons.

---

> ### Comment · Reviewer_3APB · 2025-08-05
>
> I sincerely appreciate the authors’ detailed response to my questions. However, I remain unconvinced that the proposed approach introduces a fundamentally new perspective. While the authors frame the concept of *feature–classifier mismatch* as a novel problem, many prior works in personalized federated learning (PFL) have naturally addressed similar issues—though perhaps under different terminology. In particular, methods like FedBABU, MAML-based approaches, and FedRep all incorporate mechanisms that implicitly or explicitly align features and classifiers at the client level.
>
> For instance, FedBABU separates the model into a shared feature extractor and a local classifier, updating only the extractor during training and personalizing the head on each client. This design directly targets the alignment between client-specific features and the classifier, effectively resolving what the authors now describe as a mismatch. Similarly, MAML-based methods such as Per-FedAvg allow rapid adaptation of the global model to local data, which inherently reduces any initial misalignment between the shared model and client-specific distributions. FedRep also optimizes the classifier locally before updating the global feature extractor, again implicitly correcting mismatches between features and classifiers.
>
> Moreover, even simple baselines like FedAvg with local classifier fine-tuning demonstrate substantial improvements in personalization performance, highlighting that the challenge often lies in aligning the classifier with well-learned features. In light of this, I believe the *feature–classifier mismatch* is not a newly identified problem, but rather a reframing of a widely recognized challenge. The true contribution of FedPFT lies in its specific solution—prompt-based feature transformation—rather than in the novelty of the problem itself.
>
> That said, I find the proposed methodology itself to be a promising direction. However, the benefits gained from the Prompt-based Feature Transformation (PFT) appear to be offset by the additional communication and model complexity it introduces. As a result, the practical advantage of FedPFT over simpler adaptation-based methods remains unclear. To justify a higher rating, I believe the authors need to provide stronger theoretical justification for why PFT offers a fundamentally superior solution compared to existing personalization approaches.

---

> > ### Author Response · Authors · 2025-08-06
> >
> > We sincerely thank the reviewer for the detailed follow-up and thoughtful analysis. We appreciate your recognition of the effectiveness of our prompt-based feature transformation, and we understand your concern regarding the novelty of the feature–classifier mismatch formulation.
> >
> > We agree that several existing PFL methods (e.g., FedRep, FedPer, FedBABU) can partially alleviate what we describe as mismatch. This point is explicitly discussed in the Introduction and Related Work of our paper. However, these approaches neither formally define this phenomenon nor quantitatively measure it, and their optimization objectives were not explicitly designed to eliminate it. In contrast, our work (i) provides a formal definition of feature–classifier mismatch, (ii) introduces a quantitative metric to measure it, and (iii) demonstrates through this metric that mismatch persists even in strong baselines such as FedBABU and FedPer.
> >
> > Regarding local classifier fine-tuning, we note that its strong personalization performance further highlights the importance of addressing mismatch. However, such post-training adjustments inherently leave mismatch unresolved throughout training, allowing gradient conflicts and feature extractor degradation to accumulate, as we discussed in the Introduction. Our dual-phase strategy resolves mismatch before model updates in each round, which reduces inter-client gradient divergence (Fig. 1) and yields consistent improvements—up to 5.07% over SOTA—across diverse settings.
> >
> > In summary, while prior methods may implicitly mitigate mismatch, they neither explicitly recognize nor fully address it. **Our contribution lies in** (1) making mismatch an explicit and measurable concept in PFL, and (2) designing the first training-time mechanism specifically targeting it, achieving both theoretical clarity and practical gains.
> >
> > We appreciate this opportunity to clarify our contribution, and we will revise the paper to more clearly position our work along these lines.

---

> ### Comment · Reviewer_3APB · 2025-08-06
>
> Based on the authors' final clarification—specifically, the explicit formulation of the feature–classifier mismatch and the training-time mechanism designed to address it—I am inclined to raise my score slightly in recognition of the paper’s contribution. However, I still have some reservations.
>
> In particular, it remains unclear whether the observed performance improvements stem primarily from the proposed mismatch correction mechanism or simply from increased model complexity, which could introduce additional computational and communication overhead. Moreover, if the approach relies on learning client-specific transformation modules, it is important to consider how the model would generalize to previously unseen users without local adaptation.
>
> That said, the authors have addressed many of my concerns to a reasonable extent, and I appreciate their thoughtful response. As such, I have decided to increase my score slightly (based on the assumption that authors will update the contribution more clearly).

---

> > ### Author Response · Authors · 2025-08-07
> >
> > We sincerely thank the reviewer for the thoughtful follow-up and for increasing the score. We truly appreciate your recognition of our clarification regarding the formalization and training-time resolution of the feature–classifier mismatch. Your support at this stage is extremely valuable to us. We also commit to incorporating all the improvements discussed during the rebuttal and discussion phases into the final version of the paper.
> >
> > Regarding your remaining concerns:
> >
> > **1. On performance gains vs. model complexity:**
> >
> > As discussed in *Section 4.4: Disentangling Model Capacity and Training Strategy Effects*, we specifically examined whether the observed performance improvements stem from increased model capacity or from our training design. The results show that the gains are attributable to our training-time alignment strategy, rather than model size. We will emphasize this point more clearly in the revision.
> >
> > **2. On generalization to unseen clients:**
> >
> > Two potential strategies for addressing this are as follows:
> >
> > (1) Zero-shot: The server may maintain a pool of lightweight personalized prompts from prior clients and send them to the new client. The new client can select the most suitable one by performing a single inference pass on its local training data.
> >
> > (2) Few-shot: Since FedPFT already provides a well-trained global feature extractor, transformation module, and classifier, the new client can simply initialize a prompt and adapt it with a few local updates to align local features with the classifier.
> >
> > We thank the reviewer for raising this important point. Generalization to unseen clients is indeed a valuable research problem, and we plan to explore this further—along with OOD evaluation—as part of our future work.

---

### Official Review · Reviewer_zsuL · 2025-07-03

**Clarity:** 3
**Significance:** 2
**Originality:** 3
**Rating:** 4
**Confidence:** 4

**Summary:**

This paper addresses data heterogeneity in federated learning by proposing a novel personalized federated learning framework. Unlike existing methods that localize or fine-tune certain model parameters, the proposed FedPFT framework identifies the mismatch between local features and the global classifier as the main performance bottleneck. It uses personalized prompts combined with local features, processed through a shared self-attention transformation module, to align them with the global classifier during training. Experimental results demonstrate that FedPFT outperforms state-of-the-art methods by up to 7.08%.

**Questions:**

1. The advantage of the proposed prompt-driven feature transformation over existing personalized federated learning methods based on feature alignment and classifier collaboration (e.g., FedPAC [1]) remains unclear to me. As shown in Figure 3, if $(p_{\kappa,i}, \tau_i)$ and $h_{\kappa}$ are regarded as a unified "classifier" module, the proposed framework essentially also performs global feature alignment and local classifier collaboration, similar to FedPAC. Therefore, the core advantage and key contribution of the proposed framework are not clearly differentiated.

**Ethical Concerns:**

["NO or VERY MINOR ethics concerns only"]

**Final Justification:**

The authors' clarifications and additional experimental results has effectively addressed most of my concerns. After carefully considering the authors' responses and other reviews' comments, I have decided to raise my original rating to 4.

**Limitations:**

**Limitations:**

1. The advantage of the proposed prompt-driven feature transformation over existing personalized federated learning methods based on feature alignment and classifier collaboration (e.g., FedPAC [1]) remains unclear to me. As shown in Figure 3, if $(p_{\kappa,i}, \tau_i)$ and $h_{\kappa}$ are regarded as a unified "classifier" module, the proposed framework essentially also performs global feature alignment and local classifier collaboration, similar to FedPAC. Therefore, the core advantage and key contribution of the proposed framework are not clearly differentiated.

**Paper Formatting Concerns:**

No.

**Quality:**

2

**Strengths And Weaknesses:**

**Strengths:**

1. This paper is well-written and easy to follow. Exploiting a prompt-driven transformation module to achieve alignment between local features and the global classifier for improved personalization in federated learning is both interesting and promising.

2. The transformation experiments effectively explain the motivations and insights behind the approach, making them clear and accessible to readers.

3. Extensive experiments validate the effectiveness of the proposed method across various datasets, demonstrating its superiority over existing competitors.

**Weaknesses:**

1. The advantage of the proposed prompt-driven feature transformation over existing personalized federated learning methods based on feature alignment and classifier collaboration (e.g., FedPAC [1]) remains unclear to me. As shown in Figure 3, if $(p_{\kappa,i}, \tau_i)$ and $h_{\kappa}$ are regarded as a unified "classifier" module, the proposed framework essentially also performs global feature alignment and local classifier collaboration, similar to FedPAC. Therefore, the core advantage and key contribution of the proposed framework are not clearly differentiated.

2. The evaluation section can be improved to strengthen the applicability of the proposed method to real-world scenarios:

   **(1) Scalability:** The current evaluation only considers 40 clients. The performance with a larger number of participating clients should be assessed to demonstrate scalability.

   **(2) Participation mode:** Only the full participation setting is considered in the experiments. In practice, partial participation is more common in federated learning systems and should be evaluated.

   **(3) Local update epochs:** The impact of the number of local update epochs should be evaluated to demonstrate the robustness of the proposed algorithm.

   I would be glad to increase my rating if you can validate the scalability and robustness of the proposed method, particularly with a large number of participating clients and under partial participation settings.

---

> ### Author Rebuttal · Authors · 2025-07-31
>
> We sincerely thank the reviewer for valuable feedback. We try our best to carefully address each of the raised weaknesses and questions. Additionally, we would like to emphasize that the code for our method was submitted along with the original submission, and we commit to open-sourcing the additional experiments conducted during the rebuttal phase if this paper is accepted.
>
> ## Weakness 1 & Question 1 & Limitation 1: Regarding the distinction from FedPAC
>
> We appreciate the reviewer’s thoughtful comparison. We would like to clarify that our method differs significantly from FedPAC in terms of **objective, methodological design, and practical advantage**.
>
> ### 1. Difference in Objective
>
> - **FedPAC aims to align features across clients**, using global prototypes to reduce inter-client feature divergence and performing classifier collaboration via weighted aggregation of personalized heads.
> - In contrast, **our method aims to align local features and the shared global classifier within each client**. This is a new problem that has not been studied in federated learning but is critical for stable training and accurate inference under heterogeneous data.
>
> ### 2. Differences in Methodological Design and Our Advantage
>
> While our method, like FedPAC, enables feature alignment and classifier collaboration, **it further addresses the alignment between local features and the global classifier**, **which FedPAC is not designed to handle**:
>
> - Before local training, we freeze the feature extractor and classifier, and optimize lightweight, client-specific prompts to **first resolve the mismatch** between local features and the global classifier (see Section 3.5). This ensures that training starts from an aligned feature space and avoids mutual interference between modules during local updates.
> - Although FedPAC promotes inter-client alignment, it **does not address the feature–classifier mismatch within each client**, which we identify as a core bottleneck. As shown in the table below, FedPAC still suffers from a noticeable gap between Match Accuracy and Origin Accuracy, indicating residual mismatch.
> - In contrast, our method **resolves this mismatch** and **implicitly enables inter-client feature alignment** by aligning local features to a shared semantic space defined by the global classifier. This leads to class-wise feature consistency across clients, as visualized in Appendix Fig. 9(e), where features from different clients cluster naturally by class.
>
> These results suggest that **FedPFT not only achieves inter-client consistency comparable to methods explicitly designed for it, but also addresses intra-client mismatch effectively**.
>
> |        | Dir 0.5 (Match Acc.) | Dir 0.5 (Origin Acc.) | Dir 1.0 (Match Acc.) | Dir 1.0 (Origin Acc.) |
> | ------ | -------------------- | --------------------- | -------------------- | --------------------- |
> | FedPAC | 70.79                | 69.59                 | 67.12                | 66.14                 |
>
>
>
> ## Weakness 2: Evaluation Scope – Scalability, Partial Participation, and Local Epochs
>
> We thank the reviewer for this thoughtful suggestion. In response, we have conducted additional experiments to address these concerns and summarize our findings below.
>
> ### (1) Scalability to more clients
>
> As part of our original submission, we have already included experiments involving **100 clients**, comparing FedPFT with multiple state-of-the-art baselines under varying non-IID settings. The detailed results are provided in **Appendix K, Table 12**, and for the reviewer’s convenience, we summarize them below. As shown, FedPFT consistently outperforms competing methods, even as the number of clients increases, demonstrating strong scalability.
>
> | Scenarios    | FedPer | FedBN | FedRoD | FedCAC | FedPFT    |
> | ------------ | ------ | ----- | ------ | ------ | --------- |
> | $\alpha=0.1$ | 84.68  | 85.51 | 87.58  | 87.40  | **88.43** |
> | $\alpha=0.5$ | 71.40  | 70.85 | 75.23  | 72.82  | **76.49** |
> | $\alpha=1.0$ | 66.92  | 67.18 | 70.99  | 69.06  | **72.53** |
>
> In the rebuttal phase, we additionally conducted experiments with 200 clients. As shown below, our method consistently outperforms SOTA methods.
>
> | 200 clients  | FedAvg-FT | FedBABU-FT | FedRoD | FedPFT |
> | ------------ | --------- | ---------- | ------ | ------ |
> | $\alpha=0.1$ | 86.20     | 87.04      | 85.68  | 87.90  |
> | $\alpha=0.5$ | 72.82     | 73.80      | 72.60  | 74.81  |
> | $\alpha=1.0$ | 70.04     | 71.31      | 69.80  | 72.32  |
>
> ### (2) Participation Mode
>
> We have also evaluated **partial client participation** in our original submission (see **Section 4.7, Table 7**), under **90%, 70%, and 50%** participation rates on multiple datasets. For ease of reference, we summarize those results here. The results indicate that FedPFT maintains strong robustness and performance under varying participation ratios.
>
> | Datasets  | 90%            | 70%            | 50%            |
> | --------- | -------------- | -------------- | -------------- |
> | CIFAR-10  | 73.88$\pm$1.84 | 74.21$\pm$1.45 | 74.33$\pm$1.38 |
> | CIFAR-100 | 45.74$\pm$0.32 | 45.46$\pm$1.14 | 45.87$\pm$0.81 |
> | Tiny      | 28.53$\pm$0.62 | 29.24$\pm$0.16 | 29.90$\pm$0.10 |
>
> To further test under a more challenging setting, we conducted new experiments during the rebuttal phase with only **20%** client participation, and compared against SOTA methods. As shown in the table below, FedPFT continues to achieve superior performance across all settings.
>
> | CIFAR100, Dir 0.5 | 90%       | 70%       | 50%       | 20%       |
> | ----------------- | --------- | --------- | --------- | --------- |
> | FedAvg-FT         | 41.50     | 40.88     | 41.95     | 40.23     |
> | FedBABU-FT        | 40.33     | 41.28     | 41.65     | 38.80     |
> | FedRoD            | 39.75     | 40.55     | 38.37     | 38.23     |
> | **FedPFT**        | **45.45** | **46.08** | **46.22** | **46.10** |
>
> ### (3) Impact of Local Update Epochs
>
> In the rebuttal phase, we extended our evaluation to explore the impact of increasing local update epochs ($E \in {10, 15, 20}$). As expected, larger $E$ values lead to performance degradation across all methods due to exacerbated client drift under non-IID settings. However, **FedPFT exhibits smaller performance drops**, highlighting its robustness.
>
> This robustness stems from our two-phase training design: by resolving the feature–classifier mismatch before model updates, FedPFT ensures that training begins from an aligned feature space and **mitigates interference between the feature extractor and classifier**, thus reducing the adverse effects of drift.
>
> | CIFAR10, Dir 1.0 | 10        | 15        | 20        |
> | ---------------- | --------- | --------- | --------- |
> | FedAvg-FT        | 67.73     | 67.55     | 67.23     |
> | FedBABU-FT       | 68.30     | 68.13     | 67.85     |
> | FedRoD           | 67.23     | 67.05     | 66.80     |
> | **FedPFT**       | **69.43** | **69.32** | **69.10** |
>
> We will include all of the above results and analyses in the revised version of the paper. Thank you again for the helpful and constructive suggestions.

---

> > ### Comment · Reviewer_zsuL · 2025-08-06
> > **Response to the Rebuttal**
> >
> > I sincerely appreciate the authors' clarifications and additional experimental results. Their rebuttal has effectively addressed most of my concerns. Accordingly, I would like to increase my original rating to 4.

---

> > > ### Author Response · Authors · 2025-08-06
> > >
> > > We sincerely thank the reviewer for the constructive feedback and the willingness to reconsider the rating. We truly appreciate your recognition of our clarifications and additional experiments, and we are grateful for your support in improving the quality and clarity of our work.

---

> ### Comment · Area_Chair_LV8a · 2025-08-06
> **Official Comment from AC**
>
> Dear Reviewer,
>
> As the Area Chair handling this submission, I would like to express my appreciation for the thorough and insightful comments you have provided. The authors have now submitted a comprehensive rebuttal addressing each of the points raised in your reviews.
> To facilitate a fair and rigorous decision-making process, I kindly urge each reviewer to carefully assess whether the authors' rebuttal adequately addresses your specific concerns and technical questions. Your active participation and constructive input during the upcoming discussion phase will be crucial in arriving at a well-informed decision.
>
> Please feel free to share any additional thoughts or clarifications you may have as we proceed with our deliberations.
>
> Best regards,
>
> Your AC

---

### Decision · Program_Chairs · 2025-09-17

**Decision:**

Accept (poster)

**Comment:**

This paper presents FedPFT, a personalized federated learning framework that addresses data heterogeneity by mitigating the mismatch between local features and the global classifier. The work makes valuable contributions: 1) it offers a targeted solution to feature-classifier mismatch in federated learning; 2) it introduces a practical prompt-driven framework with strong empirical performance; and 3) it demonstrates compatibility with auxiliary objectives like contrastive learning, enhancing its versatility. Key concerns raised by reviewers were effectively addressed during the rebuttal process. Thus, it is strongly recommended that the authors integrate the constructive comments from reviewers to further enhance the work’s quality.